# Spatial organization of the mouse retina at single cell resolution by MERFISH

Jongsu Choi[1,4], Jin Li[2,4], Salma Ferdous[2], Qingnan Liang[1], Jeffrey R. Moffitt ®[3] & Rui Chen ®[1,2] ✉

The visual signal processing in the retina requires the precise organization of diverse neuronal types working in concert. While single-cell omics studies have identified more than 120 different neuronal subtypes in the mouse retina, little is known about their spatial organization. Here, we generated the single-cell spatial atlas of the mouse retina using multiplexed error-robust fluorescence in situ hybridization (MERFISH). We profiled over 390,000 cells and identified all major cell types and nearly all subtypes through the integration with reference single-cell RNA sequencing (scRNA-seq) data. Our spatial atlas allowed simultaneous examination of nearly all cell subtypes in the retina, revealing 8 previously unknown displaced amacrine cell subtypes and establishing the connection between the molecular classification of many cell subtypes and their spatial arrangement. Furthermore, we identified spatially dependent differential gene expression between subtypes, suggesting the possibility of functional tuning of neuronal types based on location.

The retina is a highly organized tissue that captures and processes light signals before relaying the information to the visual cortex of the brain. This complex function is achieved through coordinated action among diverse types of neurons, each with distinct characteristics. Five major neuronal cell types exist in the retina: photoreceptor cells (rod and cone), amacrine cells (ACs), bipolar cells (BCs), horizontal cells (HCs), and retinal ganglion cells (RGCs), which can be further classified into many subtypes based on their distinct morphology, function, and gene expression. In recent years, single-cell transcriptomics studies have identified 128 distinct neuronal subtypes in the mouse retina[1–3]; however, it remains largely unknown how the large number of subtypes is spatially organized.

The characteristic laminar organization of the major retinal cell types is critical for the light signal procession in the retina, which is facilitated by synaptic connections in the plexiform layers[4,5]. Thorough investigations of the plexiform layers have characterized unique dendrite and axon patterns of certain neuronal subtypes in the specific sublaminae[6,7]. In contrast, the spatial organization of neuronal somas at the subtype level has not been systematically investigated. Previous studies indicate that some cell subtypes may also follow a general laminar organization. For example, the two AC subgroups, GABAergic and glycinergic ACs, show distinct sublayer localization within the inner nuclear layer (INL) with GABAergic ACs basally positioned closer to the ganglion cell layer (GCL) in general[8]. Furthermore, previous studies have suggested that the laminar position of cell somas could affect its function and neurites remodeling of interacting cells[9–11].

In addition to the laminar distribution, diverse cell types must also disperse across the tissue to ensure proper coverage of the vision field. While the cellular distribution in the retina allows maximum visual coverage, variations within the cell type composition and density exist across the retina. For example, cone photoreceptors expressing different opsins exist in gradients across the dorsal-ventral axis of the mouse retina with the S-opsins enriched in the ventral retina[12]. The spatial preference of a cell type such as the S-opsin cone photoceptors in the upper visual field likely stems from an evolutionary adaptation that is advantageous to survival[12]. Downstream of cone photoceptors, several bipolar cells are also shown to possess spatial preference such as the OFF subtypes decreased in the dorsal region and s-cone bipolar cells increased in the ventral/nasal regions[12,13].

[1]Department of Biochemistry and Molecular Biology, Baylor College of Medicine, Houston, TX 77030, USA. [2]Department of Molecular and Human Genetics, Baylor College of Medicine, Houston, TX 77030, USA. [3]Program in Cellular and Molecular Medicine, Boston Children's Hospital; Department of Microbiology, Harvard Medical School, Boston, MA 02115, USA. [4]These authors contributed equally: Jongsu Choi and Jin Li. ✉e-mail: ruichen@bcm.edu

Although recent progress in single cell transcriptomics technologies has generated a near complete molecular map for all cell subtypes in the retina, the current droplet-based methods require dissociation of the tissue into single cells prior to profiling, resulting in the loss of spatial information[14,15]. Given that a precise organization of cells contributes to proper synaptic connections for the circuitry and function of the central nervous system[9–11], charting the organization and interaction among neurons is critical. So far, the spatial information of cell subtypes in the retina has primarily been generated based on antibody staining and in situ hybridization of cell type specific markers. Due to the large number of cell types, coupled with the lack of truly unique specific markers for most cell subtypes, these traditional

approaches are not scalable or feasible to generate a complete spatial map. Using a recently developed imaging-based spatial transcriptomics method, MERFISH[16–22], we sought to investigate the intricate organization of different cell types and subtypes in the retina by establishing the spatial map of the mouse retina at single cell resolution (Fig. 1a). Through integration with single-cell RNA sequencing (scRNA-seq) reference data, we annotated over 100 cell subtypes and identified striking laminar organization of the retinal cell subtypes. By examining the dorsal/ventral and temporal/nasal regions of the retinal sections, we also investigated the cellular distribution patterns. Multimodal integration of MERFISH and scRNA-seq data further allowed for imputation of the entire transcriptome and revealed differential

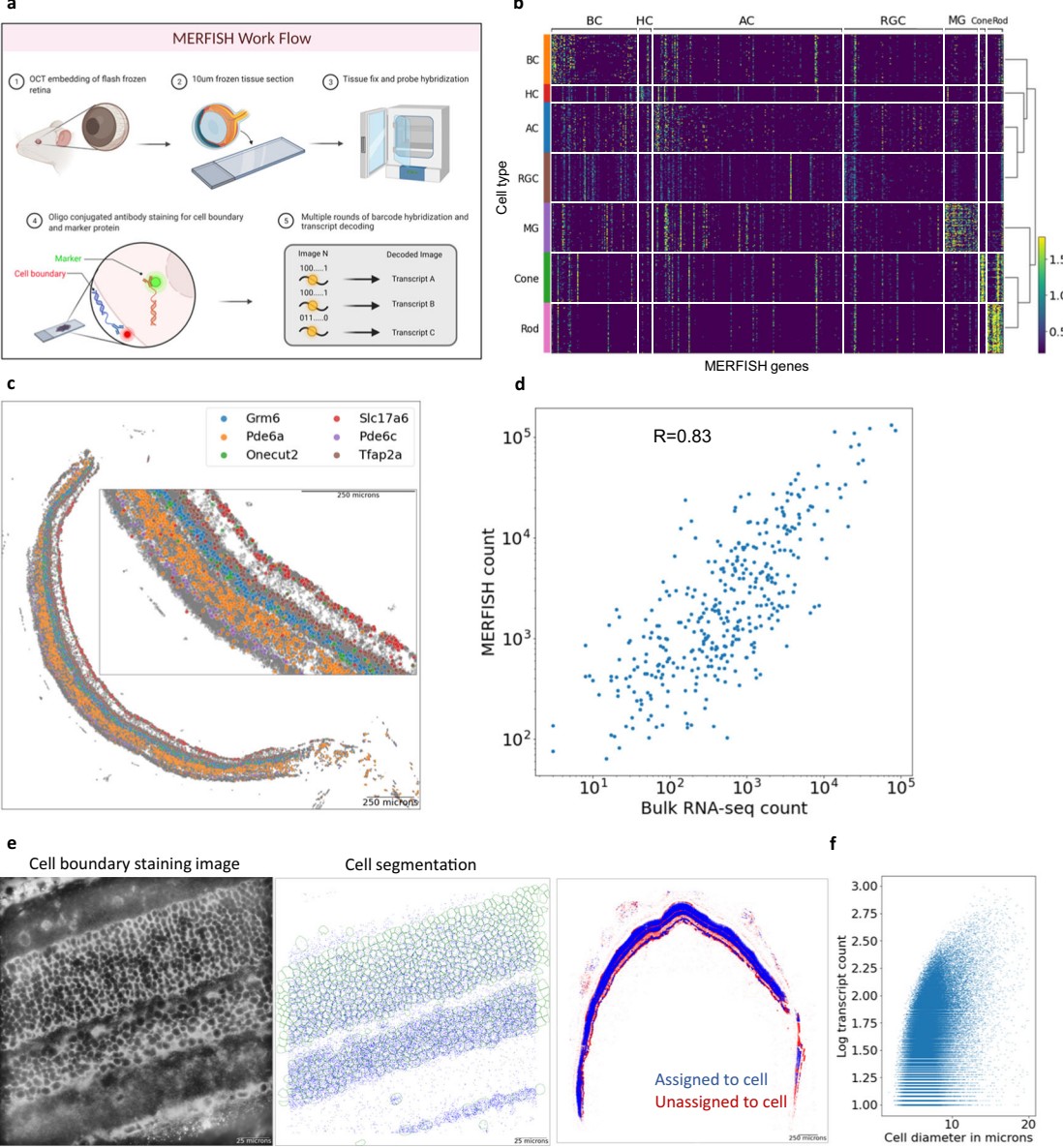

**Fig. 1 | Overview of spatial transcriptome profiling in the mouse retina.**
**a** Overview of the MERFISH protocol workflow. MERFISH experiment was conducted on 10-μm-thick mouse retina cross-sections. Cell membrane staining and in-situ hybridization of transcripts were imaged through multiple rounds of readout hybridization. **b** Heatmap of MERFISH probe transcript expression in reference scRNA-seq data. A total of 368 markers specific for major cell types and subtypes were selected from scRNA-seq data. **c** Distribution of major cell type markers in the retina tissue. Decoded transcripts show expected patterns across the tissue such as photoreceptor markers Pde6a and Pde6c in the ONL,

interneuron markers Grm6, Tfap2a, and Onecut2 in the INL, and ganglion cell marker Slc17a6 in the GCL. A local density filter was applied to remove likely false positive transcripts. **d** Transcript count comparison between MERFISH and bulk RNA-seq. A high correlation shows MERFISH transcription detection level is comparable to RNA-seq. **e** Cell boundary staining image and segmentation result. The representative field of view shows typical staining and cell segmentation.
**f** Scatter plot of cell diameter and transcript counts per each MERFISH cells. The single-cell QC metrics shows ~80 transcript counts per cell with ~7-μm diameter on average. **a** was created with BioRender.com.

gene expression among the same AC subtypes located between the INL and GCL, highlighting the importance of spatial information in cell type classification. In summary, the spatially resolved single cell reference map of the mouse retina generated in this study serves as a valuable resource for the entire vision science community and lays the foundation for many future studies, including development, cell-cell interaction, and circuitry.

## Result

### Spatial transcriptome profiling of the mouse retina

We designed a panel of 368 genes specific for major cell types and subtypes based on published and in-house generated scRNA-seq data[1–3] to capture the cell heterogeneity of the mouse retina (Fig. 1b, Supplementary Data 1). We performed MERFISH experiments using this panel on 21 wild-type C57B1/6 J mouse retinal cross-sections to generate single-cell spatial transcriptomics profiles (Fig. 1a). To investigate spatial distribution patterns of retinal cells, we performed additional MERFISH experiments on 21 wild-type cross-sections each positioned in dorsal/ventral and temporal/nasal orientations using an improved panel design of 500 genes[23] (Supplementary Fig. 1a, Supplementary Data 1). Several lines of evidence indicate the high quality of our MERFISH dataset. First, the total copy number of individual genes identified in tissue sections and experimental replicates showed high correlation, indicating excellent reproducibility (Supplementary Fig. 1a). Second, we observed proper spatial distribution of known cell type marker genes, such as photoreceptor markers *Pde6a* and *Pde6c* in the outer nuclear layer (ONL), BC and AC markers *Grm6* and *Tfap2a* in the INL and RGC marker *Slc17a6* in the GCL (Fig. 1c). Finally, the total transcript count obtained in our MERFISH experiments exhibited high correlation with the gene expression levels measured by bulk RNA-seq (Fig. 1d).

Proper cell segmentation is critical for generating high-quality single-cell spatial transcriptomics profiles. This is particularly challenging in the retina as retinal cells are densely packed with variable size and morphology. As a result, nuclei staining alone, as utilized in previous MERFISH studies, is not sufficient to precisely identify the boundary of individual cells in the retina. To solve this issue, cell membrane co-staining using oligo-conjugated antibodies was performed along with the gene panel. As shown in Fig. 1e, the intensity of cell boundary staining varies across different layers with stronger staining observed at the RGC layer, making cell segmentation challenging. To achieve optimal segmentation for cells across all retinal layers, we combined two deep-learning segmentation algorithms[24,25], whose performance varied in different retinal layers, to determine individual cell boundaries (Fig. 1e, Supplementary Fig. 2a, b, c). Upon segmentation, we obtained a total of ~390,000 cells with an average diameter of around 7 μm, which corresponds well with immunohistochemistry images (Fig. 1f). The mean number of assigned transcripts per cell was around 80 (Fig. 1f).

Integrative clustering analysis[26,27] of MERFISH single-cell transcriptome profiles identified 38 retinal clusters and 8 non-retinal clusters (Fig. 2a, Supplementary Fig. 3a). The retinal single-cell clusters were annotated as one of the six major cell types based on marker expressions such as *Pde6a* for rods, *Pde6c* for cones, *Vsx2* for BCs, *Pax6* for ACs, *Vim* for MGs, *Onecut1* for HCs, and *Slc17a6* for RGCs (Fig. 2b). After excluding photoreceptor cells, we performed sub-clustering analysis to examine non-retinal cells, which resulted in 3 non-neuronal clusters. Because our panel design did not include many variable genes expressed in non-neuronal cells, the transcript numbers in non-neuronal clusters were much lower in comparison to retinal cells (Supplementary Fig. 3b). Two non-neuronal clusters were labeled as muscle cells and oligodendrocytes by their differentially expressed genes such as tropomyosin 2, a muscle-specific protein and their specific spatial localizations outside of the retina and in the optic nerve (Supplementary Fig. 3c–d). The other non-neuronal

cluster was labeled as microglia and astrocytes based on their localization in the INL and GCL as well as the expression of *Glul* (Supplementary Fig. 3c–d). The proportion of major cell types was largely consistent with the known composition of the mouse retina with rod cells comprising around 60% of the total population and other interneuron cell types, cone, and RGC subsequently trailing (Fig. 2c). The relative lower number of photoreceptors cells[4] identified in our study may be attributed to the use of 10 μm sections, which contains multiple rod layers, smaller photoreceptor cell size, and the design of our gene panel primarily targeting interneurons and ganglion cells. The cell type annotation was confirmed by back-plotting the cell coordinates, which demonstrated proper layering patterns of the retina (Fig. 2d).

### Subtype classification through integration with scRNA-seq data

Bipolar cells identified in the clustering analysis could be grouped into three major bipolar cell types, ON rod, ON cone, and OFF cone types by the confined expression of respective markers *Prkca*, *Grm6*, and *Grik1* (Supplementary Fig. 4a). Our probe panel was not sufficient to clearly cluster and separate all BC subtypes (Supplementary Fig. 4b). To achieve better resolution in subtype annotation, we leveraged scRNA-seq data to perform co-embedding and integration[28] and identified all 15 BC subtypes in our MERFISH data (Fig. 3a, Supplementary Fig 3b, c and Methods). Each cluster of annotated BC subtypes exhibited clear and exclusive expression of known subtype markers (Fig. 3b). In addition, the resulting cell subtype proportion was relatively similar to previous estimates based on scRNA-seq data[1] with a few differences that can be attributed to the cell type enrichment method[29] used in the scRNA-seq study (Fig. 3c).

Similarly, annotated amacrine cell clusters also exhibited confined expression of canonical GABA and glycine neurotransmitter markers *Slc6a1* and *Slc6a9* (Supplementary Fig. 4d). Through co-embedding analysis with scRNA-seq reference data[3], we achieved a significantly higher resolution map (Methods), allowing for subsequent cluster annotation to their corresponding subtypes (Fig. 3d, Supplementary Fig. 4e). Our MERFISH AC subtypes showed consistent expression patterns of known subtype markers[3] (Fig. 3e). As the established AC subtype markers largely consisted of differentially expressed genes identified in scRNA-seq clusters[3] and have not been experimentally validated, our observations provide the confirmation to many subtype markers. Furthermore, overlaying the subtype annotation from the co-embedding result on the lower dimensional space calculated using only MERFISH features showed confined sub-structures of subtype labeling, providing further confidence to our prediction (Supplementary Fig. 3f). The population abundance of AC subtypes ranged from 0.006-10.26%, largely consistent with the scRNA-seq data[3] with a few notable differences (Fig. 3f). In fact, the proportion of starburst subtype (AC17) identified in our study was ~5.5% of all ACs, which is closer to the 5.2% estimate based on whole-mount staining experiment[4] compared with ~2% reported in the scRNA-seq study[3] with enrichment method.

We annotated 45 known retinal ganglion cell subtypes through co-embedding with scRNA-seq reference data[2] (Fig. 3g). Examination of the known combinatory RGC subtype markers[2] indicated that our RGC subtype annotation exhibits comparable subtype-specific marker expression (Fig. 3h). The proportion of annotated RGC subtypes in our dataset was reasonably correlated with previous estimates[2] (Fig. 3i). The annotated RGCs showed expected localization exclusively in the GCL (Supplementary Fig. 4g).

### Laminar organization of the neuronal subtypes in the retina

Retinal circuitry requires the laminar organization of major retinal cell types[30]; however, it is not well understood to what extent the same principle extends on the subtype level. Previous studies on specific retinal cell subtypes have mainly relied on antibody labeling and in-situ

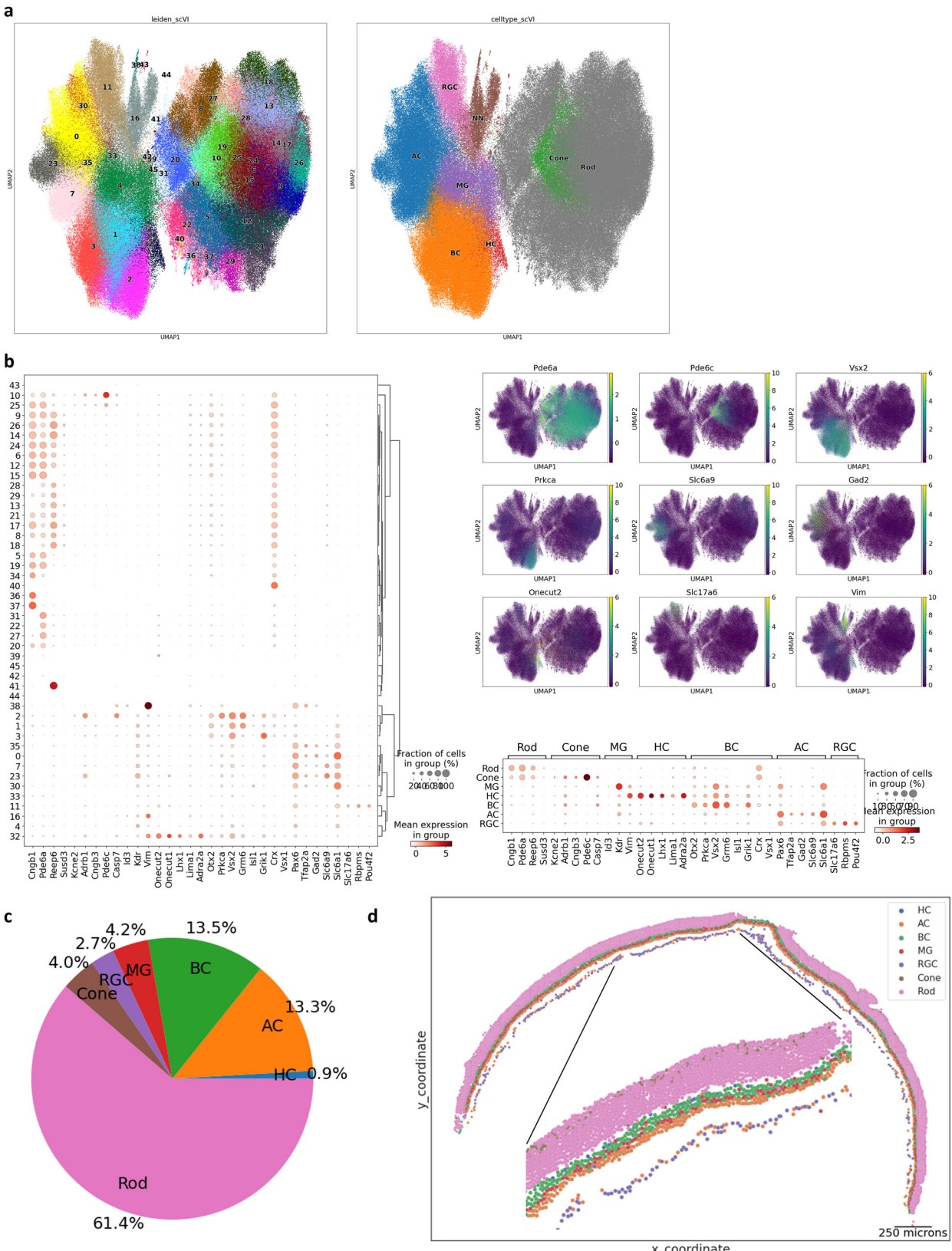

hybridization[31], which are limited to a few targets at a time. Thus, no systematic studies have been done to examine the spatial relationship among a large number of subtypes simultaneously. Using our established spatial atlas, we investigated the positioning patterns of inter-neuron subtypes by determining the cell position along the normalized INL. As expected, we observed BCs distributed around the apical half of the INL. On the subtype level, rod ON BC (RBCs) exhibited the most apical cell body relative to other BC subtypes, positioned within the top 20% of the total INL length on average (Fig. 4a, Supplementary Fig. 5a, *p*-value 0.001). In addition, RBCs showed a strong proximity enrichment with HCs, the most apical cell type in the INL, which agrees with previous observations[32,33] (Supplementary Fig. 5a).

**Fig. 2 | Major cell type identification of MERFISH single-cell profiles.**
**a** Visualization of integrated MERFISH single-cell clusters by UMAP. Integrated clustering analysis of ten MERFISH experiments resulted in 38 distinct retinal clusters, which have been annotated as major retinal cell types and 8 clusters, which have been annotated as non-major retinal cell types. **b** Major cell type marker expression in retinal single-cell clusters. Cell type-specific markers show distinct and exclusive expressions in each cluster. The following transcripts were used to annotate cell type: Rod - Cngb1, Pde6a, Reep6, and Susd3; Cone - Kcne2, Adrb1, Cngb3, and Pde6c; BC - Otx2, Vsx2, Isl1, and Grik1; AC - Pax6, Slc32a1, Tfap2a, Gad2, and Slc6a9; RGC - Slc17a6, Rbpms, and Pou4f2; HC - Onecut1, Onecut2, and Lhx1; MG - Id3, Kdr, and Vim. **c** Composition of major cell type annotation. The ratio of major cell types matches known cell type composition. **d** Tissue plot showing annotated major cell type. Major cell types can be found in appropriate retinal layers.

In contrast, BC1B, a rare population of OFF BC subtype[1,34], showed the largest distance away from the apical boundary of the INL relative to other BC subtypes with the average position of more than 50% along the total INL length (Fig. 4a, Supplementary Fig. 5b, *p*-value 0.001). Located toward the middle INL, BC1B showed a high proximity enrichment with MGs (Supplementary Fig. 5b). The intriguing preferential lamination pattern of BC1B is in agreement with previous reports of BC1B having amacrine-likeness in morphology and positions[1,34]. While we found no difference between the average ON and OFF cone bipolar cell positions (Supplementary Fig. 5c), each BC subtype showed overlapping yet distinct sublaminar distribution (Fig. 4a). Specifically, certain subtypes such as BC9 and BC6 exhibited more apical positioning, while subtypes like BC5C and BC1A showed more basal positioning. The birth timing of cone BCs precedes RBCs in general[35] and the OFF subtypes are born significantly earlier than the ON subtypes[36]. Our observation of intermixed BC subtype spacing in the INL suggests that the laminar organization is not entirely determined by the birth order of BC subtypes and a significant migration occurs during or after subtype genesis.

More than a 4-fold higher number of AC subtypes exists compared with BCs. Furthermore, the morphology or physiological functions of the majority of molecularly distinct 63 AC subtypes remain unknown. Our comprehensive annotation of AC subtypes revealed distinct positioning of each subtype in the distal INL (Fig. 4b). We observed that glycinergic subtypes were among the most apically positioned subtype whereas GABAergic subtypes were more basal, consistent with previous reports based on broadly separated AC subtypes[8,37]. Interestingly, non-GABAergic and non-glycinergic (nGnG) subtypes (AC10, AC24, and AC30), which share transcriptional similarity with glycinergic subtypes[3], were among the most apically positioned subtypes. Our observation of apically positioned nGnG subtypes is consistent with previous Ebf immunostaining reports of GlyT- ACs[8], which is specifically expressed in the 3 nGnG subtypes, in the central sublayer of the INL. A significant number of cells outside of the INL were also observed, which represent displaced ACs in the GCL[4,38] (Fig. 4b).

### Identification of 12 displaced amacrine cell subtypes
About half of the GCL is composed of ACs out of their usual location in the INL, named displaced ACs[4,38]. Several displaced ACs such as starburst, CRH+, VIP+, nGnG, and nNOS+ types have been identified based on their morphologies and marker labeling[39–44]; however, the complete list of AC subtypes that are preferentially displaced in GCL remain unknown due to their complexity. Based on our MERFISH result, all cells including ACs can be mapped to their location on the native tissues. Thus, all displaced AC subtypes in the GCL can be readily identified, allowing the calculation of the displacement ratio for each AC subtype. We identified 12 AC subtypes with significant displacement ratios ranging from 20% to 85% (Fig. 4c, d, *p*-value < 0.01). Starburst AC was the most abundant displaced AC type and accounted for nearly 20% of all ACs in the GCL, consistent with previous observations based on whole-mount experiments[4,45,46] (Fig. 4c). In addition, CRH1 (AC37), nGnG-4 (AC36), and nNOS (AC48 and AC54) were also identified as displaced subtypes, confirming previous reports[39,41,44]. Out of the 12 displaced ACs, 7 subtypes (AC2, AC7, AC21, AC32, AC39, AC44, and AC46) have not been previously

studied or reported as displaced ACs to our knowledge. Interestingly, all displaced subtypes ACs with the exception of AC36 (nGnG-4) were GABAergic. To validate subtype-specific displacement, we performed RNA in-situ hybridization[47] against molecular markers specific to the displaced AC subtypes in conjunction with pan-AC marker *Slc32a1* and pan-RGC marker *Slc17a6*. Consistent with MERFISH results, we observed cells positive for AC subtype markers and *Slc32a1*, but negative for *Slc17a6*[48] in both the INL and RGC layer, confirming their displacement (Fig. 4e).

### Asymmetrical distribution of neuronal subtypes in retinal quadrants
The proper coverage of the visual field is achieved by diverse neuronal types that are distributed across the retina tissue[49–54], yet regional variations in the distribution pattern exists for species-specific features such as the fovea in human. In mouse, a sub-population of cone photoreceptors exclusively expressing S-opsin are enriched in the ventral region with a few of the cone BC subtypes also showing similar dorsal-ventral patterns[12,13]. This unique distribution pattern in mouse is believed to provide advantageous benefits in the upper vision field for predator detection[12]. To investigate the cellular distribution patterns in different regions of the retina, we performed MERFISH experiments on 21 retinal sections each positioned in dorsal-ventral and temporal-nasal orientations and examined the cellular distribution patterns across the four retinal regions. At the major cell type level, we did not observe any obvious difference in distribution, except for a subtle enrichment of RGCs in the ventral region[55] (Supplementary Fig. 6a, b). Within BCs, we observed a significant decrease in overall OFF BC subclass in the dorsal region as previously reported[13] (Fig. 5a). Interestingly, we identified a decrease in the RBC ratio within BC population from ~35% to ~30% in the ventral region while no difference was seen between the temporal and nasal regions (Fig. 5a). To validate the dorsal/ventral asymmetry in RBC density, we performed immunofluorescence staining against Prkca, a well-known RBC marker in cross-sections and similarly observed a relative ~8% decrease in RBC numbers in the ventral region compared with the dorsal region (Fig. 5b, c). No significant difference in RBC numbers between temporal and nasal regions was observed. (Fig. 5b, c).

Some of the known subtype distribution patterns were observed in our data such as the relative dorsal depletion in individual OFF BC subtypes (BC1A, BC2, BC4)[13] as well as the temporal enrichment of 42_AlphaOFF-S RGC subtype[56] (Fig. 5d, Supplementary Fig. 6c). In addition to the known patterns, we further observed many trends and several statistically significant patterns that have not been previously described. For example, a dorsal enrichment in 16_ooDS_DV RGC subtype (dorsal and ventral preferring ON OFF direction-selective) and the nasal enrichment in 7_Novel RGC subtype were observed (Supplementary Fig. 6c) Although no obvious distribution pattern was observed in ACs, we found preferential distribution patterns across the temporal and nasal regions in three displaced AC subtypes (AC21, AC44, AC48) that are specific to the GCL (Fig. 5e). While the cross-sections used in our study limits the power to investigate cellular distributions typically done in retinal wholemounts, the distribution patterns observed in our study establishes the spatial description of molecularly classified retinal cell identities, of which many have not been previously reported.

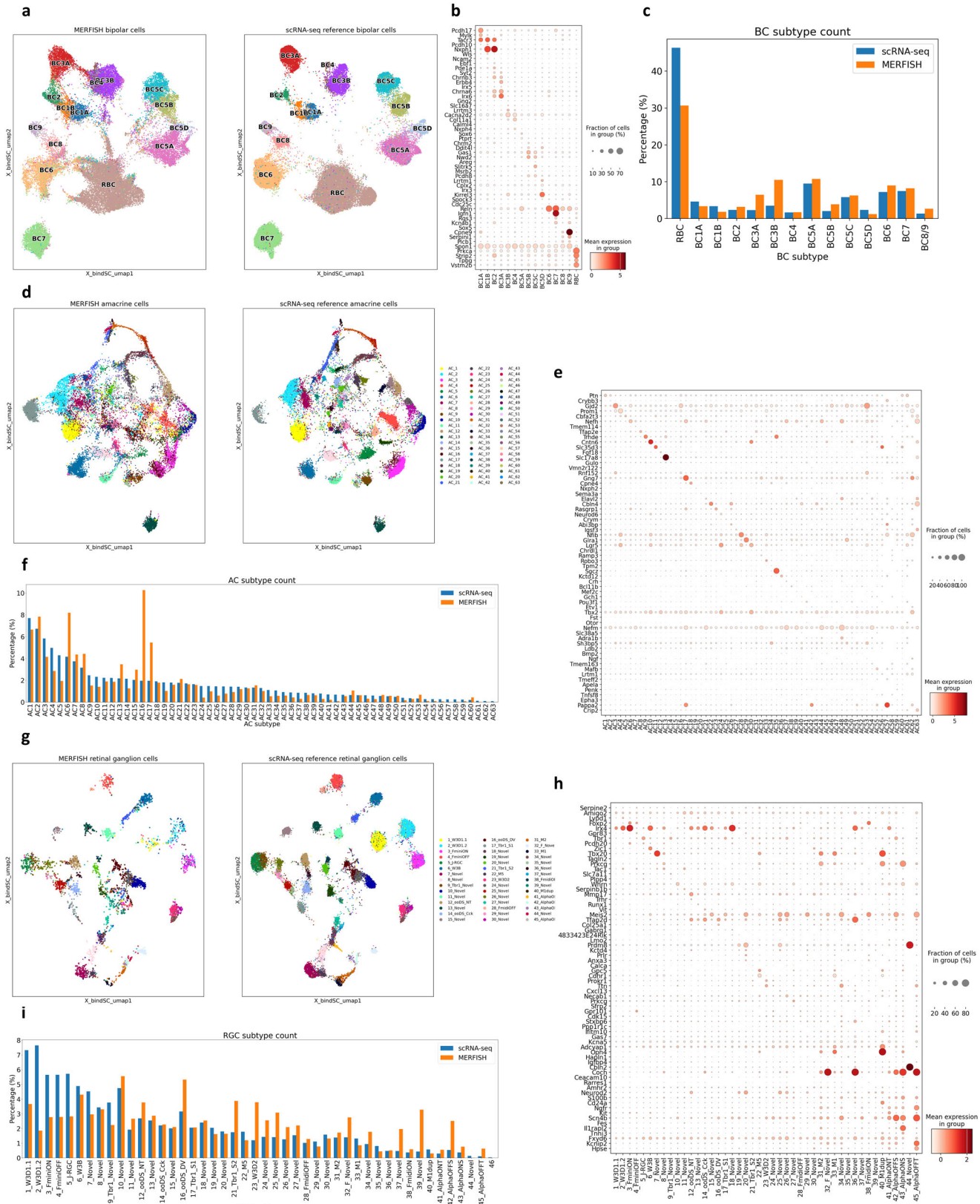

## Spatial map of the whole transcriptome through imputation

In addition to the spatial information of each cell subtype, it is highly desired to obtain spatial profiling at the individual gene level. The full spatial transcriptome can enable the systematic characterization of novel transcriptomic insights into cell type organization and cell-cell interactions. As an in-situ based spatial transcriptomics method,

MERFISH experiments with a larger gene panel will cost the overall measurement time and accuracy. Thus, we sought to resolve the genome-wide spatial profiling through a computational approach[19]. To obtain the spatial information for the entire transcriptome, we performed an imputation of the whole transcriptome by taking advantage of the co-embedding maps we generated between MERFISH and

**Fig. 3 | Subtype identification of MERFISH cells by integration with scRNA-seq reference. a** Co-embedding plot of bipolar cells between MERFISH and scRNA-seq. A high-resolution clustering map is obtained by integration between MERFISH (left) and scRNA-seq (right) with strong overlap. **b** Dot plot of bipolar cell subtype marker expression in annotated MERFISH subtypes. MERFISH BC subtypes show clear expression of known subtype markers. **c** Bar plot of annotated bipolar cell subtype ratio. The ratio of annotated BC subtypes show similarity with the published scRNA-seq study. **d** Co-embedding plot of amacrine cells. Integration between MERFISH (left) and scRNA-seq (right) ACs show reasonable overlap and separation from each subtype cluster. **e** Dot plot of AC subtype marker expression. Annotated AC subtypes show clear expression of known subtype markers. **f** Bar plot of annotated AC subtypes ratio. The ratio of annotated AC subtype is largely consistent with published scRNA-seq studies with a couple of exceptions such as the relatively high number of starburst amacrine cells in the MERFISH experiment. **g** Co-embedding plot of retinal ganglion cells. Integration between MERFISH (left) and scRNA-seq (right) RGCs shows strong overlap. **h** Dot plot of RGC subtype marker expression. Annotated RGC subtypes show clear expression of known subtype markers. **i** Bar plot of annotated RGC subtype ratio. The ratio of annotated RGC is comparable with the published scRNA-seq study.

scRNA-seq data. Upon co-embedding of MERFISH with scRNA-seq data, imputed gene expression of each MERFISH cell was calculated from weighted expression values of its neighboring scRNA-seq cells in the low dimension space[55] (see Methods for details). As showcased in Fig. 6a, the UMAP visualization of cell clusters constructed by the imputed gene expression showed expected major cell types as well as subtypes. To further evaluate the results, we examined major cell type markers in raw MERFISH and imputed transcriptomes and observed appropriate expression patterns in cell clusters and tissue location of cells (Fig. 6b, Supplementary Fig. 7a). In addition, known cell type markers not included in the MERFISH panel such as *Rho* for rods, *Crx* for rods, cones, and BCs, *Apoe* for MGs, and *Scgn* in certain BC subtypes also showed proper tissue location (Fig. 6c). Furthermore, a high positive correlation was observed between MERFISH and imputed gene expression across ten MERFISH experiments (Fig. 6d, 0.6 PCC). Lastly, we examined the Pearson correlation of cell type-specific gene expression across all cells in raw MERFISH and imputed expression profiles, which showed high correspondence (Fig. 6e). Taken together, we generated a highly accurate spatial transcriptomic map through imputation by integrating scRNA-seq with MERFISH, which can be explored to reveal new biological insights.

## Gene expression is influenced by the location of the cell

Although a very powerful approach, classifying a cell type entirely based on its transcriptomic profile has its limitation as the information of cellular physiology or morphology of the cell type is not considered. As an example, Starburst amacrine cells (SAC) can be classified into ON and OFF types based on their distinct functions and locations in the GCL and INL, respectively[56]. While a distinct transcriptional profile is observed between ON and OFF SACs during maturation, the transcriptional difference diminishes with age and the two types of SACs cannot be distinguished via scRNA-seq by P18[57]. In contrast, displaced SACs can be readily identified by MERFISH, allowing us to test if there are any subtle gene expression differences between ON and OFF SACs. Consistent with the previous scRNA-seq results, SACs found in the INL and GCL in our dataset were also mapped to only one cluster based on the imputed transcriptome, indicating their overall transcriptome profile is similar (Fig. 7a). Surprisingly, when differentially expressed gene (DEG) analysis was performed between SACs located in the INL and GCL, many genes previously known to be expressed specifically in ON and OFF SACs were identified (Fig. 7b, Supplementary Data 2). For example, *Fezf1* and *Cntn5*, which are reported to be key transcription regulator and surface protein that modulate the homophilic interaction with certain RGCs, were enriched in SACs located in the GCL (Fig. 7b)[57]. Within SACs in the INL, genes such as *Rnd3*, *Zfhx3*, and *Tenm3* were enriched, which support proper OFF SAC function through neurite modulation (Fig. 7b)[57].

We further extended our DEG analysis to all displaced AC subtypes located between the INL and GCL and identified a significant number of DEGs in AC2, AC7 and AC21 that potentially contribute to differential cell localizations (Supplementary Fig 8a). To investigate common factors driving AC displacement, we identified several consistent genes specific to the INL and the GCL across all displaced AC

subtypes (Fig. 7c, Supplementary Data 2). Within ACs found in the INL, we observed increased expressions of two transcription factors *Tcf4* and *Neurod2*, which show specific expression in glycinergic and nGnG types[3,58] as well as Cntn members. Given the preferential apical patterning of glycinergic and nGnG ACs, increased expression of *Tcf4* and *Neurod2* in non-displaced ACs may be involved in the preferential cell localization fate and patterning in the INL during retinal development. Within ACs displaced in the GCL, we found enrichment of membrane associated genes such as *Cdhr1* and *Clstn2* as well as neurofilament genes. Several cell-cell adhesion related membrane proteins have been shown to mediate interactions between specific retinal neuronal types such as ON SACs and RGCs[57,59]. Therefore, the differential gene regulation in membrane associated proteins we observed may highlight the combination codes of cell-cell adhesion related proteins required in the neuronal interactions of displaced and non-displaced AC subtypes. Gene ontology analysis of the DEGs revealed several biological pathways involved in the axonogenesis, synapse and cell-cell adhesion related processes (Fig. 7d). These biological pathways could be attributed to the physical constraints provided by the cell body localization of ACs in the INL and the GCL, which likely influence dendrite projections and synaptic connections. In short, our results indicate that distinct transcriptional regulation exists within the same subtype depending on the location of the cell. This demonstrates how spatial information can be used to further divide subtypes into distinct groups with functional and transcriptional differences, which cannot be achieved through scRNA-seq alone.

## Interactive visualization of the mouse retina spatial atlas

To facilitate the visualization and access of our mouse retina spatial atlas, we developed a user-accessible database using the CELLxGENE software (http://cellatlas.research.bcm.edu)[60]. The database can be used to visualize the imputed genes on the representative retina tissues (Supplementary Fig. 9). Moreover, the database interface allows examination of pre-computed metadata including annotated major cell types and subtypes. In conclusion, our database provides a user-friendly interactive exploration of the mouse retina spatial atlas, which will serve as a valuable tool for the vision community.

## Discussion

We generated the spatially resolved single-cell atlas of the mouse retina using an adapted MERFISH protocol and addressed key technical challenges posed by the high density and heterogeneous nature of the retina tissue. Due to the unique structure of the retina, in which each layer contains a different cell type composition, segmentation algorithms based on cell membrane staining images demonstrated varied performance in different layers. Using the combination of segmentation results from two different algorithms, we identified accurate cell boundaries to generate high-quality spatial single-cell profiles. Furthermore, we performed high-resolution annotation of our spatial single-cell profiles by integration with scRNA-seq data. We identified most known neuronal subtypes in the retina, which can be as rare as 0.05% of the entire cell population. The analysis pipeline developed in this study provides framework for similar spatial studies in other systems and tissue types.

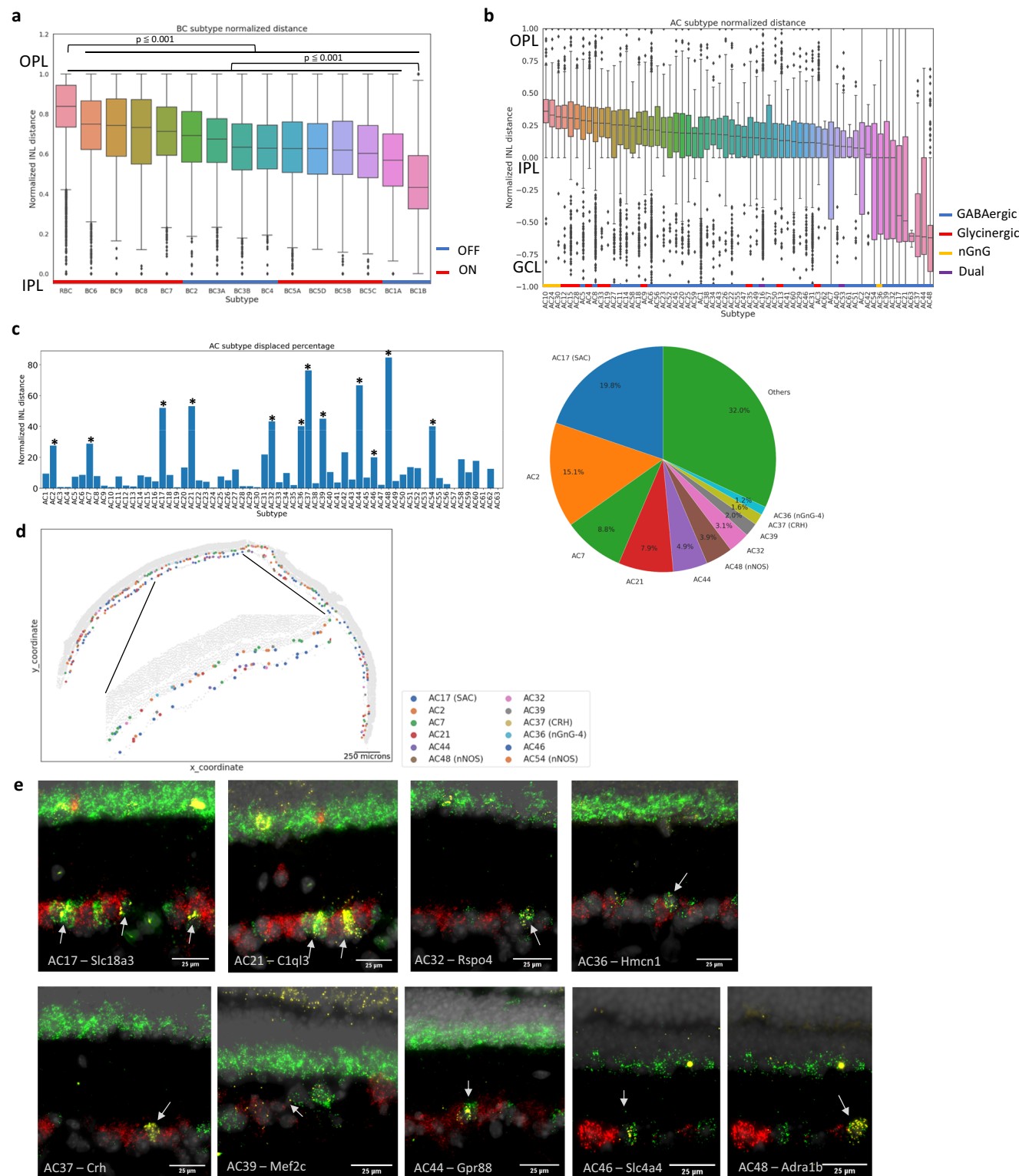

The compendium of cellular distribution generated in our study permitted us to investigate the spatial organization of nearly all retinal cell types systemically, which has previously not been possible. On a broader scale, we observed the proper distribution of major retinal cell types in the corresponding layers with a high concordance of known marker gene expression. On the subtype level, we also observed reasonable expression of known subtype markers and population composition. More interestingly, the distribution of interneuron subtypes such as BCs and ACs exhibited overlapping yet distinct positions along the retinal depth axis in the INL. Within BC subtypes, we observed

preferential localization of RBC and BC1B in the apical and central INL respectively. The spatial validation of a rare subtype such as BC1B supports the accuracy of our integration analysis and the subsequent subtype annotation. The spatial distribution of ACs demonstrated an increased population of glycinergic subtypes positioned in central INL and GABAergic subtypes in basal INL, consistent with previous lamination pattern report on broadly separated amacrine subtypes[8]. Intriguingly, three out of four nGnG subtypes, which are transcriptionally close to glycinergic subtypes[3], were among the most apically positioned subtypes. Our observation of AC laminar patterns

**Fig. 4 | Laminar organization of neuronal subtypes in the retina. a** Boxplot of bipolar subtype position in the normalized INL length. BC subtypes showed overlapping, yet distinct positioning patterns (ON types in red, and OFF types in blue). RBCs were positioned most apically compared with other subtypes (*p*-value < 0.001, pairwise *t*-test, *n* = 10), whereas BC1B showed significant basal positioning against other subtypes (*p*-value < 0.001). OFF and ON BC subtypes are marked with blue and red bars, respectively. **b** Boxplot of amacrine subtype position in the normalized INL length. Most AC subtypes showed distinct positioning within the bottom half of the INL (GABAergic in blue, glycinergic in red, nGnG in yellow, and dual in purple). Displaced AC subtypes showed an increased distribution of cells in the GCL. Glycinergic subtypes showed general apical positioning, whereas GABAergic subtypes showed more basal positioning. 3 nGnG subtypes were the most apically positioned subtypes. GABAergic, glycinergic, nGnG, and dual subtypes are marked with blue, red, yellow, and purple bars, respectively. In the box

plots, the bounds of the boxes represent 25 to 75% percentiles with the center lines showing the median. The whiskers extend 1.5 times beyond inter-quartile ranges. Individual points determined to be outliers are visualized outside of the whiskers. **c** Bar plot of displacement proportion in each amacrine subtype and a pie chart of displaced subtype composition in the GCL. Twelve AC subtypes showed significant displacement using a permutation test by shifting the subtype label for 100 times (*p*-value < 0.05, *n* = 59 sections across 10 independent experiments) and made up about 70% of all ACs in the GCL. **d** Tissue plot of displaced amacrine subtypes. Displaced AC subtypes showed distribution across both INL and GCL. **e** In-situ hybridization images of displaced AC subtype markers in the ganglion cell layer (*n* = 2). Specific markers against nine displaced AC subtypes were profiled in conjunction with pan-AC marker Slc32a1 and pan-RGC marker Slc17a6. Specific subtype markers in yellow showed co-localization with the pan-AC marker in green, but not with the pan-RGC marker in red.

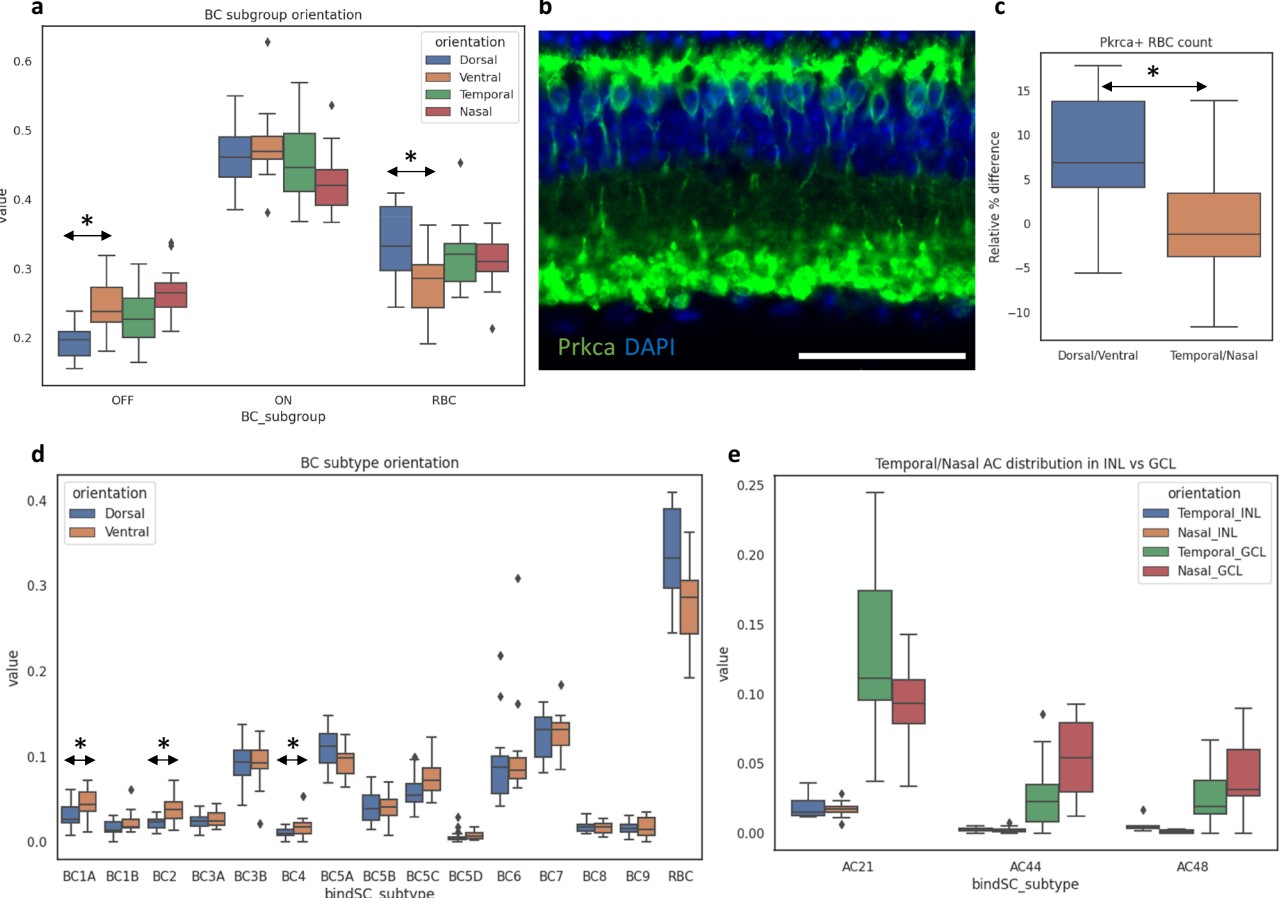

**Fig. 5 | Asymmetrical distribution of neuronal subtypes in retinal quadrants. a** Box plot of bipolar subgroup population ratio in each of the four retinal domains. A significant reduction in the number of OFF BCs was observed in the dorsal region compared to the ventral region (*p*-value < 0.0005). A significant reduction in the number of RBC was also observed in the ventral region compared with the dorsal region (*p*-value < 0.005). **b** Immunofluorescence staining against Prkca, a well-known RBC marker, in retinal cross-section (*n* = 3 each in dorsal-ventral and nasal-temporal orientations). Prkca staining; green, DAPI nuclei staining; dark blue. Scale bar: 50 μm. **c** Boxplot of relative RBC population difference quantified in the immunofluorescence staining. The overall number of RBCs in the dorsal region showed a relative -8% increase compared with RBCs in the ventral region (*p*-value < 0.0001). No significant difference in cell number was observed between the temporal versus the nasal regions. **d** Boxplot of individual bipolar subtype

population ratio in each of the four retinal domains. Some, not all, of individual OFF BC subtypes were observed show decreased cell number in the dorsal regions (*p*-value < 0.05). **e** Boxplot of three preferentially displaced AC subtype population ratio in INL and GCL across the four retinal domains. A temporal enrichment of AC21 was observed specifically in the GCL (*p*-value < 0.05, two-sample *t*-test). AC44 and AC48 showed GCL specific enrichments in the nasal retina (*p*-value < 0.0005 and *p*-value < 0.05, two-sample *t*-test). All cell type and subtype population ratios were examined over 16 nasal-temporal sections and 15 dorsal-ventral sections each across 3 independent experiments. The statistical tests were performed by two-sided Student's *t*-test. In the box plots, the bounds of the boxes represent 25 to 75% percentiles with the center lines showing the median. The whiskers extend 1.5 times beyond inter-quartile ranges. Individual points determined to be outliers are visualized outside of the whiskers.

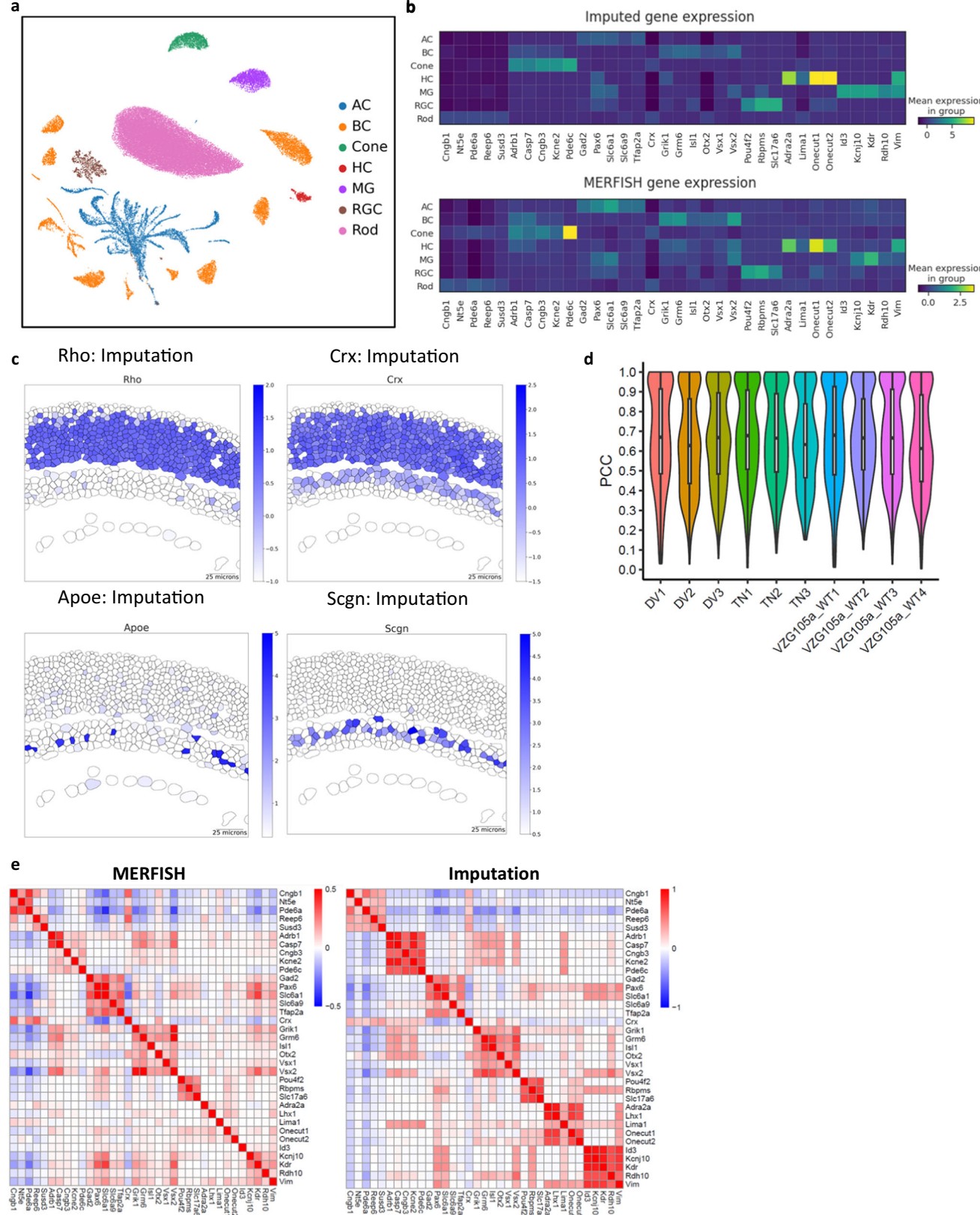

provides further molecular insight into the general basal-to-apical patterned AC subtype genesis model with earlier-born subtypes (GABAergic) distributed between the GCL and the basal INL and the later-born subtypes (glycinergic) in the central INL[8,37].

Although several AC types have been previously subjected to extensive studies, the recent molecular classification of 63 AC subtypes[3] demonstrated that most subtypes remain uncharacterized and their morphology and physiology still unknown. Our spatial atlas, which provides the connection between the molecular classification of cell types and their spatial arrangement, allows us to examine cell types with no previously available labeling method. As such, we observed 12 preferentially displaced AC subtypes with significant

**Fig. 6 | Imputation of the entire transcriptome through integration between MERFISH and scRNA-seq. a** Visualization of single-cell clusters constructed from imputed gene expression. **b** Gene expression patterns of major cell type markers in MERFISH and imputation. Both raw MERFISH and imputed transcriptome show proper expression of cell type markers in corresponding cell type clusters. **c** Inferred spatial gene expression patterns of cell type markers from imputation. Cell type-specific markers such as Rho, Crx, Apoe, and Scgn imputed from scRNA-seq data show appropriate localization. All segmented cells are included in the plot for viewing. **d** Violin plot of the correlation coefficient between MERFISH and scRNA-seq by each dataset. Positive correlation in all ten datasets indicates reliable imputation (PCC 0.65). The bounds of the boxes represent 25 to 75% percentiles with the center lines showing the median. The whiskers extend 1.5 times beyond inter-quartile ranges. **e** Correspondence between cell type marker expression between MERFISH and imputed transcriptomes. Pearson correlation coefficients between gene expression across cells show close similarities between MERFISH profiles and imputation.

---

distribution across both the INL and GCL, of which 5 have been previously reported. For example, 3 nNOS$^+$ AC types, NI, NII, and displaced cells have been identified based on varying antibody staining intensity and morphology. Our identification of 2 nNOS AC subtypes with varying displacement ratio suggests that the NII and displaced type likely match to AC52 and AC48, respectively. In addition to the subtypes previously reported to be displaced, we identified seven additional novel AC subtypes without any established key marker prior to the scRNA-seq study. The AC subtypes that are not preferentially displaced accounted for about 30% of total ACs in the GCL. It is not clear whether the displacement of ACs occurs due to the functional necessity exemplified by their axon strata or simple mis-migration during development. One possibility is that the ~30% of amacrine cells we identified in the GCL with no statistical displacement may truly have their cell bodies mis-localized in the GCL.

Variation in cellular density across the retina is driven by the biomolecular gradient and differential gene expressions during development and likely serves species specific purposes. Thus, spatial mapping of numerous neuronal subtypes in the retina is important in understanding how the complexity in cell type distribution arises during development and how their regional variation drives functional consequences. While much remains to be elucidated, our attempt at investigating subtype dispersion across the tissue confirmed some previously reported observations as well as patterns that have not been described. In fact, the reduced number of RBCs in the ventral region observed in our study was unexpected; however, as RBCs are known to form no direct connections with RGCs, the relative decreased RBC density in the ventral region may coincide well with the increased RGC density in the ventral region. Spatial transcriptomics studies with retinal wholemounts or 3-dimentional spatial omics technologies will aid in our future studies to further explore and validate the neuronal subtype distribution and variation across the retina. The transcript panel containing 368 marker genes enabled comprehensive annotation of most retinal cell types and subtypes in our study. Yet, the spatial information of the remaining transcriptome remains unknown. To expand the limited MERFISH panel, we inferred gene expression through imputation by leveraging our co-embedding integration with scRNA-seq. We provide an evaluation of our imputation through several factors. First, we observed high concordance between the imputed and measured gene expression. Second, we examined a set of known marker expressions not measured by MERFISH. Third, our cell type annotation through co-embedding integration confirms accurate spatial localization of even rare subtypes such as BC1B and known displaced AC subtypes.

While comprehensive scRNA-seq studies have nearly exhausted the cell type catalog of many tissue types including the retina, molecular classification of discrete cell types remains often challenging without taking account of features such as morphology and environmental factors. To test whether spatial information can provide insight into cell type classification, we examined displaced AC types, which possess distinct functional differences, yet cannot be distinguished into discrete clusters based on current experimental and computation methods[3,57]. Examination of the inferred transcriptome revealed many DEGs between ACs found in the INL and GCL including the ON and OFF

SACs, indicating the importance of spatial information as a factor in achieving a complete cell atlas.

## Methods

**Animal Studies:** Mouse housing, experiments, and handling were approved by the Baylor College of Medicine Institutional Animal Care and Use Committee, and the studies were conducted in adherence with the ARVO Statement for the Use of Animals in Ophthalmic and Vision Research and followed the guidance and principles of the Association for Assessment and Accreditation of Laboratory Animal Care. C57Bl/6 J mice were bred in-house and maintained in a 14 h light/ 10 h dark cyclic environment with the temperature $20 \pm 2\,°C$ and relative humidity $50 \pm 5\%$. Animals were housed by Baylor College of Medicine Center of Comparative Medicine.

Sample Collection: Mice were anesthetized with isoflurane and euthanized by cervical dislocation. Whole eyes were enucleated and immediately flash frozen on dry ice after being embedded in Tissue-TEK O.C.T. compound (VWR Cat No. 25608-930). Samples were then stored in $-80\,°C$ prior to MERFISH or RNAScope experiments. Experiments were performed using about P90 mice with no sex information (no difference in retinal tissue or cell organization between male and female mice are expected). The dorsal and temporal regions of the whole eye were marked with tissue marking dyes before enucleation for experiments that require anatomical orientation information.

MERFISH probe panel design: In designing the initial probe panel of 368 genes, we included a handful of major cell type markers for rod photoreceptors, cone photoreceptors, BCs, ACs, RGCs, HCs, and MGs. In addition, several transcription factors critical for the retina tissue function were included. To capture the cell heterogeneity on the subtype level, we included 3 to 4 top-ranked genes of each BC subtype cluster and 1–2 top-ranked genes of each AC and RGC subtype cluster identified in the publicly available scRNA-seq data[1–3]. The heatmap of the probe transcript expression was plotted using a down-sampled reference data containing 1000 of rod photoreceptors, cone photoreceptors, BCs, ACs, RGCs, MGs, and 359 HCs. The second probe panel was built on top of the initial 368 genes with a few genes replaced that showed lower than expected expression in the initial MERFISH experiments. In addition, we used SCMER (single-cell manifold-preserving feature selection) tool[23] to choose additional genes that contribute to the manifold preservation in the reference scRNA-seq data, which brought up the total number in the second panel to 500 genes.

Multiplexed Error-Robust Fluorescence in situ Hybridization (MERFISH): This protocol was adapted for fresh frozen retinas by Vizgen. In brief, 20-mm functionalized coverslips (Vizgen, #FCS01) were treated with 1% polyethyleneimine for 1 h at room temperature and washed with nuclease-free water. Coverslips were then coated with yellow green fluorescent fiducial beads (Polysciences, 17149-10) in PBS for 10 min at room temperature. The coverslip was washed twice briefly with nuclease free water and allowed to air dry. Afterwards, 10-μm-thick cryo-sections were cut near the vicinity of the optic nerve and placed onto the bead coated surface of the coverslip. The tissue sections were fixed at $-20\,°C$ with pre-chilled 100% ethanol for 30 min. The sample was brought to the benchtop and permeabilized in fresh

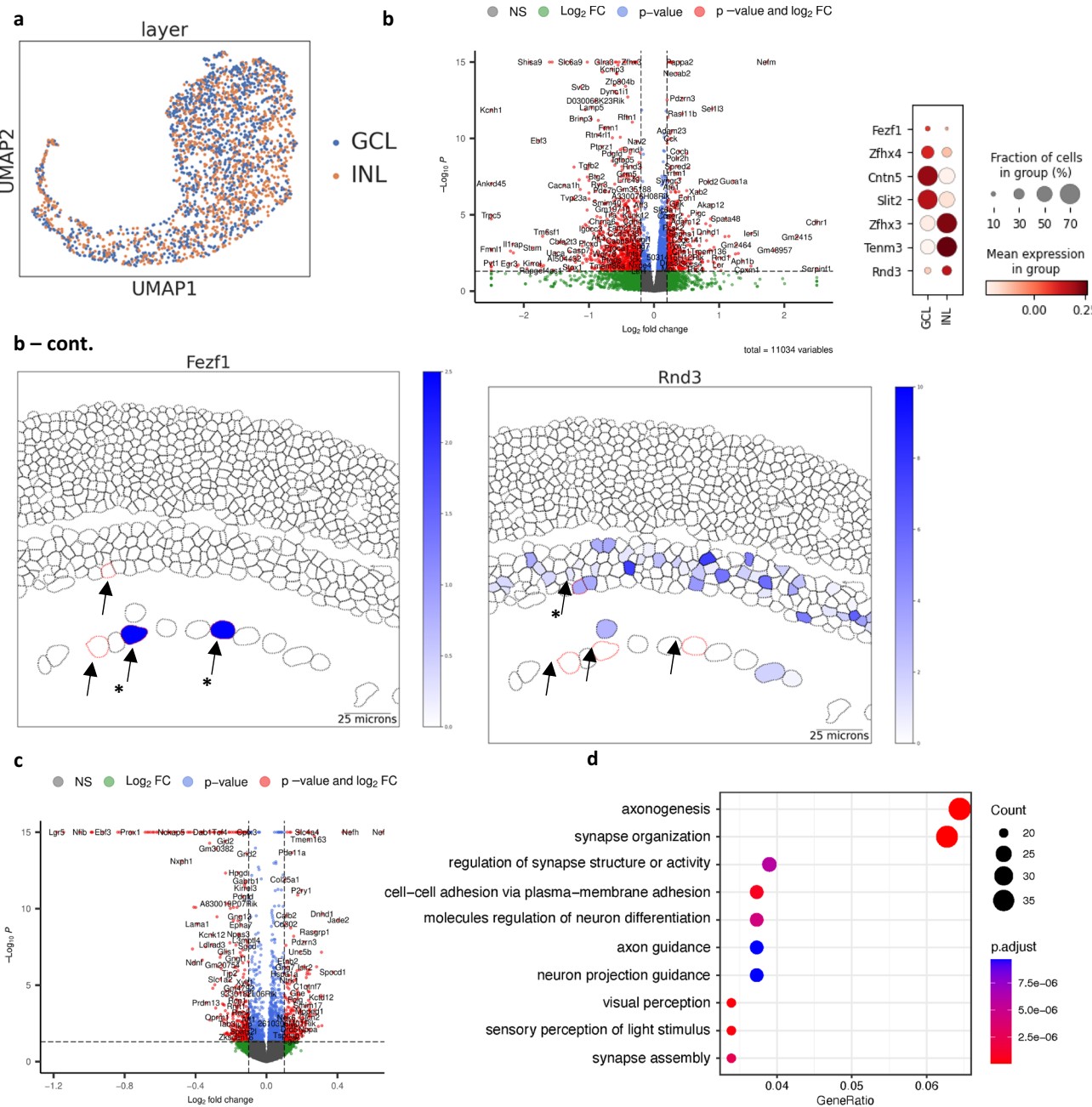

**Fig. 7 | Differential gene expression by location. a** Visualization of merged starburst amacrine cell cluster. Imputed SACs in the INL (orange) and the GCL (blue) are indistinguishable in one cluster. **b** Volcano plot of differentially expressed genes between INL and GCL starburst amacrine cells, dot plot of known location-specific genes and representative plot of Fezf1 and Rnd3 gene expression plots. Differential gene-set enrichment analysis was performed by the DESeq2 package. The p-values are calculated by the likelihood ratio test and adjusted by the Benjamini-Hochberg procedure (see methods). Several known genes expressed by ON SAC such as Fezf1, Cntn5, and Slit2 are enriched in the GCL, and genes involved in OFF SACs such as Zfh3, Tenm3, and Rnd3 are enriched in the INL. Starburst amacrine cells are marked with red outlines and arrows in INL and GCL in the tissue plot. **c** Volcano plot of differentially expressed genes in all displaced AC subtypes between INL and GCL. The consensus differential expression analysis was performed by the DESeq2 package. The p-values are calculated by the likelihood ratio test and adjusted by the Benjamini-Hochberg procedure (see methods). **d** Gene pathway analysis of differentially expressed genes in all displaced AC subtypes between INL and GCL.

room temperature 100% ethanol for 1 h. The sample was re-hydrated by sequentially exchanging buffers to 90% and 70% ethanol in a 5-minute interval and washed briefly in 2X SSC buffer. The sample was incubated for 30 min at 37 °C in 5 mL of formamide buffer before 50 μL of a custom designed probe panel containing 22 bits (Vizgen) was placed directly onto the retinal tissue. The sample was incubated with the probes in a 37 °C cell culture incubator for 36–48 h (about 2 days).

After probe hybridization was complete, the sample was washed twice with formamide buffer for 30 min each at 47 °C and then briefly

with 2X SSC three times to remove any residual formamide buffer. Cell membrane staining was done using oligo-conjugated primary and secondary antibodies provided by Vizgen (Vizgen, #CB-MM). In brief, the tissue is blocked for 1-hour at room temperature with 105-μL total volume of blocking solution (100 μL of blocking buffer plus 5 μL of murine RNAse inhibitor). Then a 1:100 primary antibody dilution was made in the blocking solution and incubated on the tissue for 1 h at room temperature. The sample was washed three times with PBST for 5 min each on a rocker and then a 1:33 secondary antibody solution was

incubated on the tissue for 1 h at room temperature. The sample was then washed three times with PBST for 5 min each on a rocker and then three times with 2X SSC. Afterwards, the sample was incubated in 50 μL of an acrylamide/bis-acrylamide based gel solution for 1.5 h at room temperature before being cleared overnight or until the tissue becomes transparent at 37 °C. After tissue clearing, the sample was washed several times in 2X SSC before the initial hybridization of fluorescence readout probes against the oligo-conjugated antibodies for 15 min at room temperature. Lastly, the sample was incubated in the wash buffer from the Vizgen Imaging Reagent Kit (Vizgen, #IK-24) for 10 min at room temperature before sequential barcode images were imaged on the Vizgen Alpha Instrument. Sequential extinguishing of the fluorescent signal and re-hybridization fluorescent read-out probes were performed by the automatic fluidic system on the Vizgen Alpha Instrument.

## Immunofluorescence

Immunofluorescent staining was performed using paraffin sections. Mice were first anesthetized with isoflurane and euthanized by cervical dislocation. The orientation of the mouse eyes was marked with tissue marking dye. The eyes were enucleated and fixed in modified Davidson's fixative overnight followed by serial dehydration steps using ethanol. Dehydrated eye tissues were cleared in xylene and embedded in paraffin. The 7 μm sections were deparaffinized, and the antigen retrieval step was performed by boiling in 10 mM sodium citrate buffer (pH 6.0) for 30 min. The slides were washed in PBS, incubated in blocking buffer (10% normal donkey serum, 0.1% Triton X-100, in PBS) for 1 h followed by incubation with primary antibody (1:1000, anti-Prkca Sigma P4334) overnight at 4 °C. On the following day, slides were washed in PBS, incubated with secondary antibody (1:100, Cy5 Donkey anti-Rabbit IgG Jackson ImmunoResearch 711-175-152) for 2 h, stained with DAPI at room temperature. The slides were mounted using anti-fade medium (Prolong; Invitrogen), and the fluorescent images were captured using a Zeiss Apotome.2 microscope (Zeiss Axio Imager). Three biological replicates in each orientation (dorsal-ventral and temporal-nasal) were performed. We counted Prkca⁺ cells in more than 7 sections for each biological replicate. The retinal region that Prkca⁺ cells belonged was separated by the middle point with equal distance to the peripheral ends in each cross-section.

## Data analysis

MERlin decoding: Raw MERFISH images were decoded by the MERlin pipeline (v0.1.9, provided by Vizgen)[61] using the 22-bit codebook designed for 368 genes in the MERFISH panel. Decoded transcripts, which represent single pixels, were exported with their barcode ID and coordinates in csv file format.

Cellpose and Mesmer cell segmentation: To detect cell boundaries in MERFISH images, cell segmentation was performed on cell boundary staining and DAPI images of individual field-of-views (FOVs). To increase relative brightness and darkness of cell membrane and DAPI staining from the background, images were adjusted by contrast enhancement using CLAHE Histogram Equalization function in OpenCV[62]. To increase the number of segmented cells per FOV, consecutive stacks Z0 and Z1 were blended by averaging intensities of the two stacks. Using the blended cell membrane images, Cellpose (v0.6.5)[25] was used to predict cell boundaries using the "cyto2" mode, which was trained on a larger dataset submitted by users. Due to the difference in compactness of the three nuclear layers that results in distinct background and staining intensities, the Cellpose generated definitive segmentation result in the ONL and INL, but not in the GCL. This may be attributed to the high signal intensity coming from the dense extracellular matrix in the GCL. To rescue non-segmented cells particularly in the GCL, we applied Mesmer[24] using "nuclear" for compartment parameter and 0.1667 for image_mpp parameter using 2-stack cell membrane and DAPI images. Mesmer resulted in near complete segmentation in the GCL, but often showed high doublet detection rate in the ONL and INL. To achieve optimal segmentation in all three retinal layers, segmented cells from Mesmer were retained if they overlapped <0.1% area of cell polygons by Cellpose. Decoded transcripts were then assigned to segmented cells by searching nearest polygons using the k-d tree algorithm in the SciPy[63]. To retain high-quality cells, segmented cells were further filtered by average DAPI intensities (>=80), radius of minimum enclosing circles (>=10, <=80), area of polygons (>=500, <=10,000), perimeter of polygons (>=50, <=400), and total transcripts (>=10).

Integration between MERFISH experiments: To reduce the batch effect between MERFISH experiments, batch correction was done to integrate different samples using scVI[26], which models the gene expression on a batch variable as well as library size and latent representation. To perform the scVI integration, raw counts of 359 genes that overlap between the two MERFISH panels were used with the "sampleID"s as the batch variable with 2 hidden layers for encoder and decoder neural networks and 30 dimensionality of the latent space. The generated 30 dimensional scVI low-representations were used to calculate a 2D UMAP for visualization of MERFISH cells by Scanpy[64]. The low-representations were also used to measure dissimilarities among cells, and the dissimilarities were used to calculate the cell clustering by the Leiden algorithm[27] with resolution 1.5.

scRNA-seq meta analysis: scRNA-seq meta analysis: The scRNA-seq reference used in subtype annotation was generated by combining publicly available scRNA-seq data for BC, AC, and RGC[1–3] and in-house single-cell/nuclei RNA-seq data. Our in-house data, which have not been published, are composed of about 130,000 cells, in which 14,000 ACs, 70,000 BCs, 400 RGCs are included. Collected datasets underwent a standardized preprocessing to exclude empty droplets, ambient RNAs, and estimated doublets (https://github.com/lijinbio/cellqc). Retained data were integrated using scVI[26] to reduce the batch effect. The trained low-dimensional representations were used to calculate the dissimilarities among cells, and cell clusters were detected by the Leiden algorithm[27]. To annotate the major cell types, top ranked genes were calculated compared to the rest of cell clusters and used to match major cell type markers. Similarly, for AC, BC, and RGC-subtyping, data integration and cell clustering were performed on corresponding subtypes, respectively. Published subtype labeling was retained and used to guide the annotation of subtype cell clusters. BC and RGC-subtyping showed exhausted subtypes. After integrating our in-house data, over-clustering of some AC subtypes annotated in public datasets were merged into single cell clusters. To retain the public subtype labels, the merged cell clusters were labelled by the subtype of the majority number of cells. For example, a merged cell cluster was labeled as AC21 while it includes subtypes AC21, AC54, and AC63 in the public datasets.

Cell identity assignment: To identify the cell type of MERFISH cells, a two-level annotation was performed for major cell type annotation and subtype annotation in sequence. To annotate the major cell type of MERFISH cells, we first calculated top ranked genes of Leiden cell clusters using Scanpy and manually inspected the known major cell type marker genes to assign major cell type identities[27,64]. Following the major cell type annotation, subtype annotation was performed by co-embedding method using scRNA-seq reference data for isolated BC, AC, and RGC subset. Using bindSC[28], scRNA-seq reference and MERFISH count matrices were bridged by optimizing correlation in the sample and feature levels simultaneously. Using bindSC, which integrates two different single-cell modalities by optimizing correlations in the sample and feature levels simultaneously, the reference scRNA-seq and MERFISH count matrices were aligned. The generated canonical correlation vectors are co-embedding low-dimensional (e.g., 15) latent representations for scRNA-seq and MERFISH cells. These co-embeddings were used to calculate a 2D UMAP between the two modalities. To annotate subtypes of MERFISH cells, the cell labeling of

scRNA-seq reference was transferred by SVM classifiers. Specifically, the latent representations of scRNA-seq cells were used to train multi-class SVM classifiers by the known scRNA-seq cell labels. Per MERFISH cell, scRNA-Seq neighbors (e.g., 3 neighbors) were detected in the co-embedding latent space, and the average of low-dimensional representations was used to represent the feature for the classification. Classification probabilities were calculated against the trained SVM classifiers, and the cell type was assigned by the maximum classification probability of the classifiers. BC, AC and RGC subtypes are annotated by applying the label transfer using the bindSC co-embedding.

Cell clustering using MERFISH genes and subtype annotation overlay: Subtype annotation from co-embedding analysis using reference scRNA-seq data was overlaid on the graphical clustering map derived from using only MERFISH transcriptome. Subsets of BCs, ACs, and RGCs were extracted from a representative MERFISH experiment ("sampleID" = "merfish_wt_JC2"). Leiden clustering was performed using Scanpy in each cell type data subset[27,64], and cells were colored by the subtype annotation labels from bingSC co-embedding.

Boundary estimation and distance calculation: The ONL and INL boundaries were estimated by approximating surrounding curves that cover the centroids of segmented cells in micrometers. To do this, the R package *alphahull*[65] was used to calculate the alpha-shape and alpha-convex hull of cell centroids. Alpha-shape was defined as an extension of convex hull, where the shape was segments connecting alpha-extreme points. A cell was called an alpha-extreme point when an open ball of radius alpha existed with the centroid on the boundary and did not cover other cells. The alpha-convex hull stored the arcs of open balls that connect neighboring alpha-extreme points. The alpha value of 100 μm generated the alpha-shape with a tight estimate of tissue boundaries including basal and apical boundaries. To separate basal and apical boundaries, the nature of an arc-shape for a retina tissue was utilized. Each section of retina tissue was assumed to have a hypothetical center. This tissue center was compared with each arc of the alpha-convex hull. Apical boundaries contained arcs that were outward away from the tissue center. On the contrary, arcs of basal boundaries were inward to the tissue center. A small number of mis-labeled boundaries were corrected interactively using the R package *shiny*[66]. To calculate distances of cells to the two estimated boundaries, perpendicular distances were calculated to each segment of boundaries using the function *nearestPointOnSegment()* in the R package *maptools*[67]. The minimum distance of segments was the calculated distance of a cell to a boundary. To normalize the thickness among tissue sections, the distance ratio was calculated by the distance to the basal distance over the sum of distances to the two boundaries.

Determining significance of cell displacement: To identify AC subtypes displaced in the GCL, the INL boundary information previously generated was used to assign ACs to their resident layer. For each subtype, the proportion of cells displaced in the GCL was calculated. To determine the significance of displacement, we applied a permutation test to calculate the expected proportion of displaced ACs by shuffling the subtype label within tissue sections for 1000 times[68]. The *p*-value was calculated as the percentage of permutations that have a greater proportion of cells in the GCL layers compared to the observed proportion. The AC subtypes with *p*-value < 0.05 were determined to be significantly displaced.

RNAScope in-situ hybridization: The RNAscope HiPlex Assay (ACD Biosystems)[47] was performed according to the ACD protocol for fresh-frozen tissue. In brief, mice were anesthetized with isoflurane and euthanized by cervical dislocation. Whole mouse eyes were then enucleated and embedded in TissueTEK O.C.T. compounds, cut in 10 μm sections and placed on glass microscope slides. After fixation with 4% FPA in PBS and permeabilization using Protease IV (RNA-Scope), the tissue sections were incubated with probes against displaced AC markers, pan-AC marker *Slc32a1* (319191-T1), and pan-RGC

marker *Slc17a6* (319171-T2). Following probes were used to stain for specific displaced AC subtypes; AC17 (448771-T5), AC21 (495681-T7), AC32 (402021-T9), AC36 (834921-T10), AC37 (316091-T3), AC39 (421011-T6), AC44 (317451-T11), AC46 (452981-T5), AC48 (413561-T6). The probes were amplified according to the manufacturer's instructions and labeled with the following fluorophores for each experiment: Alexa 488 nm, Atto 550 nm, and Atto 647 nm. The Zeiss Apotome.2 microscope (Zeiss Axio Imager) was used to visualize the FISH signals.

Calculating the periodicity of each retinal cell types and subtypes across the four retinal regions:

The coordinates of spatial single-cell profiles in each cross-section were rotated so the most posterior part of the section (optic nerve side) is at the top. Using the minimum and the maximum XY coordinates of cells, the dorsal-ventral cross-sections and the temporal-nasal cross-sections were divided in the middle (close to the optic nerve) to assign dorsal, ventral, temporal, nasal regions. The labels of four retinal regions were visually inspected to ensure the sections were divided equally. The number of each cell type and cell subtype occurrence was counted separately within the two retinal regions per each section, and the proportion was normalized by the number of all cells or within the specific cell type. Any cell type proportion difference between the two regions with *p*-value less than 0.05 was considered significant.

Tangram imputation: To resolve the entire transcriptome of MERFISH cells, we impute the gene expression by mapping single-cell RNA-Seq reference to spatial cells using Tangram[55]. Tangram calculates the imputation by summing over scRNA-seq gene expression weighted by mapping probabilities. Tangram trains the mapping probabilities by optimizing correlations between imputed expressions and MERFISH measurements of the MERFISH panel. To reduce the bias caused by scRNA-seq cells with a different cell type from a MERFISH cell, mapping probabilities of the only scRNA-seq cells with the same cell type were used for the imputation. To minimize the crossover and retain the spatial proximities of subtypes in the co-embedded space, imputation was performed within isolated AC, BC, and RGC subtypes separately with the subtype labeling annotated by bindSC co-embedding. To measure the performance of Tangram imputation, Pearson correlation coefficients have been calculated between imputed gene expressions and MERFISH measurements for the genes included in the MERFISH probe panel. A UMAP[69] was also generated using imputed gene expressions to visualize the labeling of MERFISH cells.

Gene ontology analysis: To identify any enriched biological pathways or functions in the DEGs between ACs in the INL and displaced ACs, we performed gene ontology (GO) analysis. First, gene symbols of DEGs were searched against the GO term annotation databases using the R function *enrichGO()* of the package *clusterProfiler*[70,71]. GO terms categories include biological processes, cellular components, and molecular functions. The enriched GO terms were identified under *q*-value < 0.05. To cluster GO terms that share common gene symbols, the enrichment map was generated for the enriched GO terms using the R function *emapplot()*[72].

Data resource visualization: Access to the spatial map of mouse retina is hosted at https://bcm.box.com/s/bwh011jg0m7b6t38j 3q6chsvf9xbk0wg. The data is shared in a.h5ad format with gene expression matrix and curated metadata. To provide a user-friendly interactive exploration of the spatial map, the imputed gene expression are made accessible via cellxgene[60]. The two plots contained in the data resource; the tissue plot showing the coordinates of identified MERFISH cell centers and UMAP visualization graph calculated using the imputed gene expressions. The metadata of annotated cell types can be used to color the cells in the plots. Furthermore, expression pattern of a gene or set of genes can be visualized in a histogram and in the tissue plot with the color scale as expression value. The URL of the cellxgene web service is accessible at http://cellatlas.research. bcm.edu.

Differential gene expression analysis by location: To identify the differentially expressed genes between ACs in the GCL and ACs in the INL, we adapted the workflow of *DESeq2*[73] for the imputed gene expression. First, the imputed gene counts were normalized using the function *computeSumFactors()* in the R package *scran*. Per each displaced AC subtype, the normalized values were fit into a negative binomial model by *glmGamPoi*. The likelihood-ratio test was applied to test the statistical significance by comparing with a reduced model *-1*, and the calculated *p*-values were adjusted by the Benjamini−Hochberg procedure. The log2 fold changes were also reported in the Supplementary Data 2. Differentially expressed genes were identified under *q*-value < 0.05 and |log2FoldChange | > 0.2. To calculate the shared DEGs across 12 displaced AC subtypes, a generalized linear model was used to detect the consensus DEGs across displaced AC subtypes (Full model: ~ subtype + layer; Reduced model: ~ subtype). To visualize the DEGs, the volcano plot was generated by the R package *EnhancedVolcano*[74].

### Reporting summary
Further information on research design is available in the Nature Portfolio Reporting Summary linked to this article.

## Data availability
The MERFISH data generated in this study have been deposited and are available at Zenodo (https://doi.org/10.5281/zenodo.8144355). Source data are provided with this paper.

## Code availability
The code for MERFISH image analysis is available at https://github.com/RCHENLAB/SpatialMmMERFISH and Zenodo (https://doi.org/10.5281/zenodo.8143414).

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

## Acknowledgements

This project was funded by NIH/NEI R01EY022356, R01EY018571, S10OD032189, Chan Zuckerberg Initiative (CZI) award CZF2019-002425, RRF to R.C., and NLM fellowship program T15LM007093 to S.F. We thank Vizgen for the transcript probes, the transcript decoding pipeline and the experimental guidance.

## Author contributions

Experiment conceptualization J.C., J.L., S.F., Q.L. and R.C.; MERFISH experiment J.C., and S.F.; Data analysis: J.C. and J.L.; Data resource development J.L.; RNAScope validation J.C.; Writing, reviewing and editing J.C., J.L., S.F., J.R.M., and R.C.

## Competing interests

J.R.M. declares the following competing interests. J.R.M. is a founder of, stakeholder in, and scientific advisor for Vizgen. J.R.M. is an inventor on patents associated with MERFISH applied for on his behalf by Harvard University and Boston Children's Hospital and licensed to Vizgen. The remaining authors declare no competing interests.
