## [Peer Review file · Nature Communications]

REVIEWER COMMENTS

Reviewer #1 (Remarks to the Author):

General comments:

The manuscript of Choi, et al. describes the results of MERFISH-based spatial transcriptomic analysis of adult mouse retina, with 386 different probe sets analyzed. The data appears to be generally of good quality, with high reproducibility between experimental and biological replicates, and good mappability to cell clusters identified using sc/snRNA-Seq analysis. While there are relatively few technical concerns with this study, the manuscript is generally lacking in novel biological insight. Furthermore, since the data here has much lower power to discriminate individual cell subtype than sc/snRNA-Seq (c.f. Extended Fig. 4b,f), the bulk of the manuscript simply confirms the reliability of the technology, which is all pretty well characterized in other systems. It's thus not clear it's particularly useful as a community resource either.

One of the major advantages of studying the rodent retina is its relatively homogenous organization, with individual cell types located in stereotypical and predictable positions along the radial axis. This means that few cell-specific markers are going to show expression patterns that aren't already obvious from sc/snRNA-Seq data. One of the few exceptions, which is discussed here, are displaced amacrine cells. The authors identify 9 subtypes of GABAergic and one subtype of nGnG amacrine cell as partially or (in two cases) exclusively located in the GCL. These include the well-characterized starburst amacrine subtype, but also others whose function are unknown. This observation is both interesting and novel, and can potentially give insight both into the specific function of the displaced amacrine cells and potential mechanisms mediating the formation and maintenance. However, aside from an analysis of gene expression differences in starburst amacrine cells, which basically just confirms the results of a recent SMART-Seq-based analysis from the Kolodkin and Sanes groups, no discussion or analysis of differences in expression between INL-restricted and displaced amacrine cells is included.

The manuscript needs novel insight of this sort to be appealing to a general readership. This shouldn't be too hard to do, particularly with the availability of sc/snRNA-Seq data from both the author's group and previous studies. Also unexplored, and potentially very interesting, are previously undescribed dorso-ventral, naso-temporal, or antero-posterior differences in gene expression that might persist into adulthood, as well as differences in gene expression between astrocytes in the optic nerve head and GCL.

Specific comments:

1. It is not clear to this reader why identity assignments cannot be made for the non-neuronal cells. While the top unique cell-specific markers for these cells may be absent, there ought to be enough other combinations of markers (particularly among the many transcription factors tested). This point needs further discussion.

2. Cells in the outermost 1-3 layer of the ONL do not seem to express any of the genes tested here, including canonical rod markers (c.f. Fig. 1c, 5c). This, as well as the fact that the 10 micron sections here likely included multiple rod layers, likely leads to a substantial underestimate of the relative fraction of rod cells (Fig. 2c) relative to the gold standard values reported in Jeon, et al. This point also needs discussion.

Reviewer #2 (Remarks to the Author):

Choi et al. have used a leading spatial transcriptomic method, MERFISH, to localize cell types in the mouse retina. The work seems to have been done to a high technical standard. I have no major technical concerns, but will instead discuss the results obtained. This is because, since its introduction in 2015, MERFISH has been applied to a variety of tissues including cerebral cortex (Zhang, Nature, 2021; Fang, Science, 2022), hypothalamus (Moffitt, Science, 2018; Boeshaghi, Nature, 2001; Osterhout, Nature, 2022); liver (Lu, Cell Disc, 2021) and aging brain (Allen, Cell, 2023) as well as cultured cells (Chen, Science, 2015; Moffitt, PNAS, 2016; Wang, Sci Rep, 2018; Xia PNAS 2019). Therefore, application to a new tissue is of limited general interest and the paper should be judged by its findings. Here, the novelty is limited.

Figures 1 and 2 present the method and establish its validity and reliability. Figure 3 compares MERFISH results to those from prior scRNAseq studies, showing that a large fraction of the cell types identified previously are recovered in the MERFISH dataset. Subsets of types, and proportions of types seen by MERFISH are also largely consistent with the scRNAseq results. This is excellent replication but of limited novelty.

Figure 4 describes the laminar distribution of cell types, again largely confirming previous results. For example, rod bipolar cells are grouped near the OPL, as shown many years ago by Wassle, Taylor and others. Overlapping distribution of ON and OFF cone bipolars is also consistent with prior work from their groups and others. Differential distribution of individual types are documented (Figure 4A), but the differences are small and it seems that the position of only a single type is significantly different from the other 13. In other words, this result supports the previously described intermixing of ON and OFF types. Moreover, the outlier type (called GLUMI or BC1B) has in fact previously been shown to have unusual "non-bipolar" properties by two groups. Regarding amacrine cells, the apical location of nGnG cells is consistent with earlier work (Figure 4b), as is the differential gene expression by starburst amacrine cells in the INL and GCL (Figure 6). A potentially novel result is identification of 12 amacrine types in the GCL (Figure 4c), of which only a minority have previously been shown to have this localization. However, no biological significance is attributed to this distribution so it may be of limited general

interest. Likewise, use of imputation methods adds detail to patterns of gene expression (Figure 5), but no conclusions are drawn from this added power.

The overall problem may be that the laminar distribution of retinal cells has already been described in great detail. The most interesting and poorly understood aspect of retinal cell distribution is rather in the orthogonal plane, since the many cell types must cover the entire visual field. Phenomena such as tiling and mosaic distribution have been invoked as strategies to ensure full coverage, but some asymmetries have also been documented and there is much left to learn. One wishes that the authors had applied their elegant method to en face views, which would have allowed them to address these issues.

Reviewer #3 (Remarks to the Author):

In this manuscript, Choi et al. perform spatial transcriptomic analysis of 368 genes in the ~P90 mouse retina using MERFISH. To generate a per-cell gene count, they adapted traditional MERFISH multiplexed imaging to the high cell density of the retina by using a cell membrane labeling approach followed by deep-learning segmentation approaches tuned to each retinal layer. This was performed on 21 retinal sections derived from 4 different biological specimens. Clustering analysis revealed groups of cells with marker gene expression consistent with the principal cell types of the retina. Furthermore, these clusters were comprised of cells of all four biological specimens, and in general, the biological specimens integrated well with each other. The cell type annotations applied to the clusters correlated well with the laminar distribution of each cell type. The authors then performed co-embedding and integration of their MERFISH data with existing scRNA-seq datasets to further delineate bipolar, amacrine, and ganglion subtypes. Validation of this co-embedding/integration approach was performed by comparing the resulting subtype abundances to those determined from previous publications (bipolar subtypes, Starburst amacrine), as well as transcriptional analysis of marker genes measured by MERFISH known to be restricted to specific subtypes (amacrine and ganglion cells). In total, they identified all previously known subtypes except for a single ganglion subtype (44_Novel), which is present at low abundance and likely undersampled.

The authors then performed spatial analysis of retinal cell subtypes in the laminar (apical-basal) axis of the retina. Bipolar subtypes were found in the apical INL with RBCs being more apically positioned than ON- and OFF- cone bipolar subtypes. Amacrine cell subtypes differed in their relative laminar positions. In general, they found that glycinergic amacrine cells had a more apical position compared to more basally located GABAergic subtypes, consistent with previous reports. The authors found that ~1/2 of

amacrine cells were found outside of the INL and they calculated the proportion of each amacrine subtype that was displaced. 12 amacrine subtypes had “significant” displacement (defined by authors as ranging between 23-80% of the population located outside of the INL). The proportion of displaced Starburst amacrine cells in their MERFISH data matches previous reported. Of the 12 types of displaced amacrine cells, 8 had not been previously reported as being displaced and they confirm this with ISH using subtype specific markers as well as pan-amacrine and pan-ganglion markers.

The authors next impute the expression of the full transcriptome on their MERFISH dataset using scRNA-seq data. Imputed gene expression values showed high correlation with measured values from MERFISH. Using these imputed values, the authors perform differential expression analysis between ON and OFF Starburst amacrine cells which they find cluster together in low-dimensional space. Differential gene expression analysis of ON and OFF populations revealed differential expression of genes that support subtype specific features. The authors extend this analysis to all displaced amacrine subtypes and find differentially expressed genes between GCL- and INL-positioned cells of several amacrine subtypes.

Finally, the authors created an online platform with graphical user interface to facilitate exploration of their data. The platform allows users to query gene expression data both in uMAP space as well as in physical space across the retinal sections.

Understanding the spatial organization of cellularly complex tissues is important in understanding their function. A resource such as this that provides single cell resolution of all retinal cell types will be useful for the community. The data are of high quality and the online data portal make the resource assessable for many potential users. The manuscript is suitable for publication with minor revisions.

Minor Revisions

1. The spatial analysis performed in the manuscript is performed entirely in the laminar (apical-basal) axis of the retina. While this is important for validating cell-type designations the authors make, the resource would be more useful if it also included each cell’s relative position along the central-peripheral axis of the retina.
2. Using cell position across the central-peripheral axis of the retina, could the authors calculate the periodicity of each retinal subtype as well as the evenness of each subtype’s distribution.
3. In text, the authors say that Fig 4E includes markers Pax6 and Rbpms, but in the figure legend, they list slc32a1 and slc17a6. The authors should correct this.

Response to the reviewers

We thank the reviewer for their critical assessment of our manuscript. We address all issues point-by-point in this letter. The reviewers' comments are in blue font, and our response and corrections are given in indented blocks below.

Reviewer 1# comments:

The manuscript of Choi, et al. describes the results of MERFISH-based spatial transcriptomic analysis of adult mouse retina, with 386 different probe sets analyzed. The data appears to be generally of good quality, with high reproducibility between experimental and biological replicates, and good mappability to cell clusters identified using sc/snRNA-Seq analysis.

We thank the reviewer for the kind comments on our data.

While there are relatively few technical concerns with this study, the manuscript is generally lacking in novel biological insight. Furthermore, since the data here has much lower power to discriminate individual cell subtype than sc/snRNA-Seq (c.f. Extended Fig. 4b,f), the bulk of the manuscript simply confirms the reliability of the technology, which is all pretty well characterized in other systems. It's thus not clear it's particularly useful as a community resource either.

We thank the reviewer for this critique. While spatial transcriptomics have been applied in other systems, the application has been primarily focused on a few systems such as the cell culture and brain, in which the technical challenges have been well addressed. The densely packed retina tissue provides significant technique challenges in generating high-quality spatial transcriptomics data at true single cell resolution. Thus, the novelty of our study not only includes the confirmation and exploration of otherwise previously impossible findings, but also the bioinformatics approach we used to segment cells and annotate our spatial single-cell profiles on the high-resolution subtype level by leveraging scRNA-seq data. As a comprehensive spatial map that contains nearly all molecularly classified cell types and cell subtypes in the same tissue section, our data will provide a first established reference for anyone seeking spatial information of any given neuronal subtypes in the retina.

One of the major advantages of studying the rodent retina is its relatively homogenous organization, with individual cell types located in stereotypical and predictable positions along the radial axis. This means that few cell-specific markers are going to show expression patterns that aren't already obvious from sc/snRNA-Seq data. One of the few exceptions, which is discussed here, are displaced amacrine cells. The authors identify 9 subtypes of GABAergic and one subtype of nGnG amacrine cell as partially or (in two cases) exclusively located in the GCL. These include the well-characterized starburst amacrine subtype, but also others whose function are unknown. This observation is both interesting and novel, and can potentially give insight both into the specific function of the displaced amacrine cells and potential mechanisms mediating the formation and maintenance.

We thank the reviewer for these kind remarks on our findings. We hope our findings will shine light onto the unexplored importance of spatial information in classifying cell types.

However, aside from an analysis of gene expression differences in starburst amacrine cells, which basically just confirms the results of a recent SMART-Seq-based analysis from the Kolodkin and Sanes groups, no discussion or analysis of differences in expression between INL-restricted and displaced amacrine cells is included.

We thank the reviewer for this comment and the helpful suggestion. The spatially dependent gene expression difference we identified in Starburst amacrine cells and two additional amacrine cell subtypes is from imputed transcriptome and solely derived from co-embedding between MERFISH and reference scRNA-seq data. With less than 400 features included in our MERFISH experiment, we did not observe any significant gene expression difference in the raw MERFISH data between ACs located in the two retinal layers. As the imputed gene expression relies heavily on the quality of co-embedding, accurate identification of DEGs and meaningful biological insight will be strengthened by additional MERFISH experiments with a larger gene panel and larger scRNA-seq reference data to allow proper mapping and separation of the cells found in two retinal layers. In fact, we observed DEGs in one additional displaced AC subtype (AC7) between the INL and GCL with additional MERFISH data we generated. Our goal here was to demonstrate the power of co-embedding analysis to leverage spatial transcriptomics in mapping the subtle gene expression difference between INL and GCL ACs in scRNA-seq data.

To examine the gene regulation difference between INL and GCL restricted ACs, we further examined the list of DEGs and explored potential genes of interest such as Tcf4, Neurod2 in the INL and membrane associated proteins in the GCL in the result section on page 14. We have also added gene ontology analysis results, which showed interesting biological pathways such as axonogenesis and synaptic related processes (Fig. 7d). Spatially dependent differential gene regulation we identified through imputation will be useful for optimizing probe panel design for further investigations in the future.

The manuscript needs novel insight of this sort to be appealing to a general readership. This shouldn't be too hard to do, particularly with the availability of sc/snRNA-Seq data from both the author's group and previous studies. Also unexplored, and potentially very interesting, are previously undescribed dorso-ventral, naso-temporal, or antero-posterior differences in gene expression that might persist into adulthood, as well as differences in gene expression between astrocytes in the optic nerve head and GCL.

We thank the reviewer for the helpful suggestions and have performed additional MERFISH experiments along the dorso-ventral and naso-temporal orientations to investigate the cell type distribution patterns. We performed 3 MERFISH experiments in each orientation and added more than 40 additional retinal sections, more than double the

number of cells from the initial manuscript. We have explored the distribution patterns of cell types and cell subtypes and added interesting biological insights we have observed such as the decrease in RBC count in the ventral region, which was validated using immunofluorescent staining (shown in added Fig. 5a-c) and preferentially displaced AC subtypes with temporal/nasal distribution patterns that are specific in the GCL (shown in added Fig. 5e). The new results are added in the result section titled “Asymmetrical distribution of neuronal subtypes in retinal quadrants”.

Specific comments:

1. It is not clear to this reader why identity assignments cannot be made for the non-neuronal cells. While the top unique cell-specific markers for these cells may be absent, there ought to be enough other combinations of markers (particularly among the many transcription factors tested). This point needs further discussion.

We thank the reviewer for this suggestion and have carried out further analysis. Since our current probe panel design primarily focuses on retinal neurons and was not intended to be used in classifying non-neuronal cells, we observed significantly reduced number of transcripts in non-neuronal cells (shown in added Extended Data Figure 3d) reducing the power to decipher their cell type identifies. Thus, we performed sub-clustering analysis (shown in added Extended Data Figure 3b), examined differentially expressed genes and spatial localization and annotated them as oligodendrocyte in the optic nerve, microglia/astrocytes in the retina, and miscellaneous muscle cells outside of the retinal layer (shown in added Extended Data Figure 3c). Unfortunately, we do not have sufficient power to accurately annotate and examine interesting biological insights in microglia and astrocytes.

2. Cells in the outermost 1-3 layer of the ONL do not seem to express any of the genes tested here, including canonical rod markers (c.f. Fig. 1c, 5c). This, as well as the fact that the 10 micron sections here likely included multiple rod layers, likely leads to a substantial underestimate of the relative fraction of rod cells (Fig. 2c) relative to the gold standard values reported in Jeon, et al. This point also needs discussion.

We thank the reviewer for this comment and have discussed the potential reasons for cell type composition difference in the result section on page 6. The dense transcripts found above the Pde6c (cone marker) are outer segments in Fig. 1c. We observed a reduced number of transcripts identified closer to the outer segments in the ONL. Since our probe panel design primarily aims to distinguish the interneuron and ganglion subtype heterogeneity, photoreceptor cells in the outermost 1-3 layers have likely been filtered out due to low transcripts and the comparably small cell size in the single-cell analysis. This likely contributed to the differences in photoreceptor cell proportions compared to published literature. The plot in Fig. 5c shows the imputed gene expression of single-cell profiles with all segmented cells plotted for viewing. We have addressed this in the figure legend on page 38.

Reviewer 2# comments:

Choi et al. have used a leading spatial transcriptomic method, MERFISH, to localize cell types in the mouse retina. The work seems to have been done to a high technical standard. I have no major technical concerns, but will instead discuss the results obtained. This is because, since its introduction in 2015, MERFISH has been applied to a variety of tissues including cerebral cortex (Zhang, Nature, 2021; Fang, Science, 2022), hypothalamus (Moffitt, Science, 2018; Boeshaghi, Nature, 2001; Osterhout, Nature, 2022); liver (Lu, Cell Disc, 2021) and aging brain (Allen, Cell, 2023) as well as cultured cells (Chen, Science, 2015; Moffitt, PNAS, 2016; Wang, Sci Rep, 2018; Xia PNAS 2019). Therefore, application to a new tissue is of limited general interest and the paper should be judged by its findings. Here, the novelty is limited.

We thank the reviewer for the kind remarks. The retina tissue poses a unique challenge with its highly heterogenous cell types and heavily packed nuclei, which requires a particularly careful assignment of transcripts to individual cells. In comparison, the brain and liver cells can be segmented using nuclei or polyT staining with relative ease or do not possess high cell type complexities. Therefore, technical challenges overcome in our study to generate the comprehensive spatial atlas as well as the molecular validation of previous findings and new biological insights we observed are novelties that we believe are of interest to a general audience.

Figures 1 and 2 present the method and establish its validity and reliability. Figure 3 compares MERFISH results to those from prior scRNAseq studies, showing that a large fraction of the cell types identified previously are recovered in the MERFISH dataset. Subsets of types, and proportions of types seen by MERFISH are also largely consistent with the scRNAseq results. This is excellent replication but of limited novelty.

We thank the reviewer for the kind comments on the quality of the data. Given that our goal is to create a spatially resolved true single-cell atlas of the mouse retina, rigorous establishment of accuracy and quality of the data such as the probe design, cell segmentation, and clustering analysis is needed. Thus, we chose to allocate multiple figures to highlight the reliability and validity of our data. Our approach in leveraging the scRNA-seq data was crucial in the annotation of retinal cells on the subtype level in our MERFISH data. We hope our multi-modal integration approach will provide future applications in high-resolution cell type annotation of other challenging spatial transcriptomics data.

Figure 4 describes the laminar distribution of cell types, again largely confirming previous results. For example, rod bipolar cells are grouped near the OPL, as shown many years ago by Wassle, Taylor and others. Overlapping distribution of ON and OFF cone bipolars is also consistent with prior work from their groups and others. Differential distribution of individual types are documented (Figure 4A), but the differences are small and it seems that the position of

only a single type is significantly different from the other 13. In other words, this result supports the previously described intermixing of ON and OFF types. Moreover, the outlier type (called GLUMI or BC1B) has in fact previously been shown to have unusual “non-bipolar” properties by two groups. Regarding amacrine cells, the apical location of nGnG cells is consistent with earlier work (Figure 4b), as is the differential gene expression by starburst amacrine cells in the INL and GCL (Figure 6). A potentially novel result is identification of 12 amacrine types in the GCL (Figure 4c), of which only a minority have previously been shown to have this localization. However, no biological significance is attributed to this distribution so it may be of limited general interest. Likewise, use of imputation methods adds detail to patterns of gene expression (Figure 5), but no conclusions are drawn from this added power.

We thank the reviewer for the kind remarks and detailed summary of our findings. Our findings consistent with previous publications not only validate our technical and computational methods, but also highlight the power of our spatial map to comprehensively validate almost all molecularly classified subtypes simultaneously on tissue sections. While many retinal subtypes have been described in previous studies, many molecularly identified subtypes remain uncharacterized as in the case of most AC subtypes. Likewise, the 7 out of 12 preferentially displaced AC subtypes we identified also have not been subject to any previous studies as far as we know. Our spatial characterization of these subtypes will provide a good starting point for studying AC displacement in future studies.

The spatially dependent differential gene expression observed in Starburst amacrine cells (SAC) and three additional AC subtypes is derived from imputed transcriptome through co-embedding with scRNA-seq data and represents the power to resolve previously undetectable difference between INL ACs and GCL ACs in the merged scRNA-seq clusters. To examine the biological significance of differential gene regulation in AC displacement, we have further explored the list of shared DEGs between displaced and non-displaced ACs and added our findings such as increased *Tcf4* and *Neurod2* expression in the INL and membrane proteins in the GCL in the result section on page 14. We have also performed gene ontology analysis and added the results, which showed interesting biological pathways such as axonogenesis, synaptic related processes and cell-cell adhesion (Fig. 7d). The spatially dependent gene regulation we observed in displaced AC types will provide a molecular and spatial reference in future spatial transcriptomics and mechanistic studies on AC displacement.

The overall problem may be that the laminar distribution of retinal cells has already been described in great detail. The most interesting and poorly understood aspect of retinal cell distribution is rather in the orthogonal plane, since the many cell types must cover the entire visual field. Phenomena such as tiling and mosaic distribution have been invoked as strategies to ensure full coverage, but some asymmetries have also been documented and there is much left to

learn. One wishes that the authors had applied their elegant method to en fact views, which would have allowed them to address these issues.

We thank the reviewer for this suggestion and have performed additional MERFISH experiments to investigate cell distributions across different regions in the retina tissue. Using retinal cross-sections along the dorso-ventral and naso-temporal orientations, we have added ~270,000 single-cell profiles, more than double the number of single-cell profiles included in the initial manuscript. Some of the biological insights we observed include the reduced RBC density in the ventral region (shown in added Fig. 5a-c) and temporal/nasal distribution patterns of 3 preferentially displaced ACs in the GCL (shown in added Fig. 5e) have been added in the new result section titled “Asymmetrical distribution of neuronal subtypes in retinal quadrants”.

Additionally, we have performed spatial analysis such as the neighbor enrichment and found preferential enrichment of several homotypic subtype pairs. As an example, we identified an enrichment in BC9-BC9 pairs (figure shown below), which has been previously reported to form tight clusters¹. However, we feel the use of cross-sections in our study and the relatively low number of cells profiled compared with wholemounds limits the power to make proper conclusions. Unfortunately, due to technical barriers, we are currently unable to perform MERFISH experiments in the wholemound; however, there will be future studies on the use of retinal wholemounds and 3-dimensional spatial transcriptomics profiling.

Reviewer 3# comments:

In this manuscript, Choi et al. perform spatial transcriptomic analysis of 368 genes in the ~P90 mouse retina using MERFISH. To generate a per-cell gene count, they adapted traditional MERFISH multiplexed imaging to the high cell density of the retina by using a cell membrane

labeling approach followed by deep-learning segmentation approaches tuned to each retinal layer. This was performed on 21 retinal sections derived from 4 different biological specimens. Clustering analysis revealed groups of cells with marker gene expression consistent with the principal cell types of the retina. Furthermore, these clusters were comprised of cells of all four biological specimens, and in general, the biological specimens integrated well with each other. The cell type annotations applied to the clusters correlated well with the laminar distribution of each cell type. The authors then performed co-embedding and integration of their MERFISH data with existing scRNA-seq datasets to further delineate bipolar, amacrine, and ganglion subtypes. Validation of this co-embedding/integration approach was performed by comparing the resulting subtype abundances to those determined from previous publications (bipolar subtypes, Starburst amacrine), as well as transcriptional analysis of marker genes measured by MERFISH known to be restricted to specific subtypes (amacrine and ganglion cells). In total, the identified all previously known subtypes except for a single ganglion subtype (44_Novel), which is present at low abundance and likely undersampled.

The authors then performed spatial analysis of retinal cell subtypes in the laminar (apical-basal) axis of the retina. Bipolar subtypes were found in the apical INL with RBCs being more apically positioned than ON- and OFF- cone bipolar subtypes. Amacrine cell subtypes differed in their relative laminar positions. In general, they found that glycinergic amacrine cells had a more apical position compared to more basally located GABAergic subtypes, consistent with previous reports. The authors found that ~1/2 of amacrine cells were found outside of the INL and they calculated the proportion of each amacrine subtype that was displaced. 12 amacrine subtypes had “significant” displacement (defined by authors as ranging between 23-80% of the population located outside of the INL). The proportion of displaced Starburst amacrine cells in their MERFISH data matches previous reported. Of the 12 types of displaced amacrine cells, 8 had not been previously reported as being displaced and they confirm this with ISH using subtype specific markers as well as pan-amacrine and pan-ganglion markers.

The authors next impute the expression of the full transcriptome on their MERFISH dataset using scRNA-seq data. Imputed gene expression values showed high correlation with measured values from MERFISH. Using these imputed values, the authors perform differential expression analysis between ON and OFF Starburst amacrine cells which they find cluster together in low-dimensional space. Differential gene expression analysis of ON and OFF populations revealed differential expression of genes that support subtype specific features. The authors extend this analysis to all displaced amacrine subtypes and find differentially expressed genes between GCL- and INL-positioned cells of several amacrine subtypes.

Finally, the authors created an online platform with graphical user interface to facilitate exploration of their data. The platform allows users to query gene expression data both in uMAP space as well as in physical space across the retinal sections.

We thank the reviewer for the kind comments, the comprehensive summary and interpretation of our manuscript.

Understanding the spatial organization of cellularly complex tissues is important in understanding their function. A resource such as this that provides single cell resolution of all

retinal cell types will be useful for the community. The data are of high quality and the online data portal make the resource assessable for many potential users. The manuscript is suitable for publication with minor revisions.

We thank the reviewer for the kind remarks. We hope our data will be a valuable contribution to the field and provide a framework for future spatial transcriptomics studies.

Minor Revisions

1. The spatial analysis performed in the manuscript is performed entirely in the laminar (apical-basal) axis of the retina. While this is important for validating cell-type designations the authors make, the resource would be more useful if it also included each cell's relative position along the central-peripheral axis of the retina. 2. Using cell position across the central-peripheral axis of the retina, could the authors calculate the periodicity of each retinal subtype as well as the evenness of each subtype's distribution.

We thank the reviewer for this suggestion and have performed additional MERFISH experiments along the dorso-ventral and naso-temporal orientations to calculate the periodicity of each retinal subtype across the four retinal regions. In the added result section titled "Asymmetrical distribution of neuronal subtypes in retinal quadrants", we validated some of previously described observations and identified new findings such as the RBC density asymmetry that have been added in fig. 5 and Extended Figures Data Figure 6.

3. In text, the authors say that Fig 4E includes markers Pax6 and Rbpms, but in the figure legend, they list slc32a1 and slc17a6. The authors should correct this.

We thank the review for pointing it out and apologize for the mistake. Slc32a1 and Slc17a6 were used for pan-AC and pan-RGC markers in the RNAScope assay. The main text has been corrected.

1. Nadal-Nicolás, F. M. *et al.* True S-cones are concentrated in the ventral mouse retina and wired for color detection in the upper visual field. *eLife* **9**, e56840 (2020).

REVIEWERS' COMMENTS

Reviewer #1 (Remarks to the Author):

The authors have addressed all my major outstanding concerns, and have considerably expanded the dataset in the process, adding interesting new data demonstrating differential distribution of different neuronal subtypes along the dorsoventral and nasotemporal axes. This is a well executed study that will be of broad general interest to the retinal community.

My only remaining minor concerns are the presence of several awkward or confusing passages in the revised text, including:

Pg. 14: "that can potentially reflect the differential regulation driving cell localization"

Pg. 14: "may suggest that Tcf4 and Neurod2 are involved in cell localization fate and patterning during retinal development"

Pg. 15: "ACs the INL and the GCL"

Pg. 14: "which likely contribute to difference in dendrite projections to form synaptic connections."

Once these are fixed, the manuscript should be suitable for publication.

Reviewer #2 (Remarks to the Author):

The authors have worked hard to improve their manuscript, choosing regional differences as the avenue to novel biological insight. These and other new data substantially improve the manuscript.

Reviewer #3 (Remarks to the Author):

The authors have added additional data regarding the main criticism of the spatial distribution of cell types across the retina, i.e. within the cardinal dimensions of the retina, not just within the apical-basal plane. They provide box plots of overall distribution in 4 quadrants, nasal-temporal and dorsal-ventral.

With the data from their new sections, is it not possible for them to give the coordinates in a more granular way? i.e. distance from dorsal most margin, distance from most nasal margin, etc? This would make the additional data more valuable and would provide a more powerful resource.

Response to the reviewers

We thank the reviewers again for their critical assessment of our manuscript. We address all issues point-by-point in this letter. The reviewers' comments are in blue font, and our response and corrections are given in indented blocks below.

Reviewer 1# comments:

The authors have addressed all my major outstanding concerns, and have considerably expanded the dataset in the process, adding interesting new data demonstrating differential distribution of different neuronal subtypes along the dorsoventral and nasotemporal axes. This is a well executed study that will be of broad general interest to the retinal community.

We thank the reviewer for the helpful suggestions and kind comments on our revised manuscript.

My only remaining minor concerns are the presence of several awkward or confusing passages in the revised text, including:

Pg. 14: "that can potentially reflect the differential regulation driving cell localization"

Pg. 14: "may suggest that Tcf4 and Neurod2 are involved in cell localization fate and patterning during retinal development"

Pg. 15: "ACs the INL and the GCL"

Pg. 14: "which likely contribute to difference in dendrite projections to form synaptic connections."

Once these are fixed, the manuscript should be suitable for publication.

We thank the reviewer for the suggestions and have revised the text.

Reviewer 2# comments:

The authors have worked hard to improve their manuscript, choosing regional differences as the avenue to novel biological insight. These and other new data substantially improve the manuscript.

We thank the reviewer for the helpful suggestions and kind remarks on our revised manuscript.

Reviewer 3# comments:

The authors have added additional data regarding the main criticism of the spatial distribution of cell types across the retina, i.e. within the cardinal dimensions of the retina, not just within the apical-basal plane. They provide box plots of overall distribution in 4 quadrants, nasal-temporal and dorsal-ventral. With the data from their new sections, is it not possible for them to give the coordinates in a more granular way? i.e. distance from dorsal most margin, distance from most nasal margin, etc? This would make the additional data more valuable and would provide a more powerful resource.

We thank the reviewer for the helpful suggestions and kind comments. The high number of subtypes and the limited number of cells within individual sections make it difficult to show obvious distribution patterns and draw meaningful conclusions. Thus, we opted to examine the cellular distribution within the four larger retinal quadrants. Our limited analysis using four regions of sections instead of two showed similar results; however, we believe that the additional factors make the interpretation more difficult and do not add much new insight to our conclusion. Future studies with larger data, or 3-dimensional spatial analysis should provide more insights.